# High-spin Co³⁺ in cobalt oxyhydroxide for efficient water oxidation

Xin Zhang [1,9], Haoyin Zhong [1,9], Qi Zhang [1], Qihan Zhang [1], Chao Wu[2,3], Junchen Yu [1], Yifan Ma[1], Hang An[1], Hao Wang[1], Yiming Zou [4], Caozheng Diao [5], Jingsheng Chen [1], Zhi Gen Yu [6], Shibo Xi[2] ✉, Xiaopeng Wang [1,3,7,8] ✉ & Junmin Xue [1] ✉

Cobalt oxyhydroxide (CoOOH) is a promising catalytic material for oxygen evolution reaction (OER). In the traditional CoOOH structure, Co³⁺ exhibits a low-spin state configuration ($t_{2g}^6 e_g^0$), with electron transfer occurring in face-to-face $t_{2g}^*$ orbitals. In this work, we report the successful synthesis of high-spin state Co³⁺ CoOOH structure, by introducing coordinatively unsaturated Co atoms. As compared to the low-spin state CoOOH, electron transfer in the high-spin state CoOOH occurs in apex-to-apex $e_g^*$ orbitals, which exhibits faster electron transfer ability. As a result, the high-spin state CoOOH performs superior OER activity with an overpotential of 226 mV at 10 mA cm⁻², which is 148 mV lower than that of the low-spin state CoOOH. This work emphasizes the effect of the spin state of Co³⁺ on OER activity of CoOOH based electrocatalysts for water splitting, and thus provides a new strategy for designing highly efficient electrocatalysts.

The generation of hydrogen through electrochemical water splitting is considered a highly promising approach for harvesting energy and alleviating intermittent availability issues associated with renewable energy sources[1,2]. Nevertheless, the overall efficiency of water splitting is strikingly hampered by the sluggish kinetics involved in the anodic oxygen evolution reaction (OER), which often exhibits multiple coupled electron-proton transfer steps[3,4]. Consequently, substantial efforts have been devoted to developing OER electrocatalysts that can facilitate rapid electron transfer capabilities during the OER process[5–7]. Among the various electrocatalysts evaluated so far, cobalt oxyhydroxides (CoOOH) have been widely recognized as promising candidates due to their earth abundance, high electrochemical activity, and adjustable electronic structure[8–10].

In the conventional CoOOH structure, Co³⁺ ($3d^6$) typically adopts a low-spin state configuration ($t_{2g}^6 e_g^0$), where electron transfer occurs in face-to-face $t_{2g}^*$ orbitals. To date, most studies have mainly focused on enhancing the electron transfer abilities in the low-spin state Co³⁺ CoOOH species, via morphology engineering[10], metallic ion doping[11], oxygen vacancy introduction[12], heterojunction creation[13], etc[14]. As instructed by simulation[15], the OER activities of CoOOH could be significantly enhanced by incorporating more active high-spin state Co³⁺ ions to replace low-spin state Co³⁺ ions. Although the introduction of high-spin state Co ions has been reported in Co-based oxides[16–19], most of them would experience an irreversible reconstruction process under anodic alkaline conditions, forming CoOOH based materials, which act as the actual catalytic species for oxygen evolution[20,21]. Till now, there has been no experimental evidence showing the effect of

[1]Department of Materials Science and Engineering, National University of Singapore, Singapore 117575, Singapore. [2]Institute of Sustainability for Chemical, Energy and Environment (ISCE), Agency for Science, Technology and Research (A*STAR), Singapore 627833, Singapore. [3]College of Materials Science and Engineering, Sichuan University, Chengdu 610065, China. [4]School of Materials Science and Engineering, Nanyang Technological University, Singapore 639798, Singapore. [5]Singapore Synchrotron Light Sources (SSLS), National University of Singapore, Singapore 117603, Singapore. [6]Institute of High Performance Computing (IHPC), Agency for Science, Technology and Research (A*STAR), Singapore 138632, Singapore. [7]State Key Laboratory of Intelligent Construction and Healthy Operation and Maintenance of Deep Underground Engineering, Sichuan University, Chengdu 610065, China. [8]Tefusen Semiconductor & Hydrogen Energy Technology (Yunnan) Co., Ltd, Wenshan Zhuang and Miao Autonomous Prefecture 663200, China. [9]These authors contributed equally: Xin Zhang, Haoyin Zhong. ✉e-mail: Xi_shibo@isce2.a-star.edu.sg; msewxia@nus.edu.sg; msexuejm@nus.edu.sg

high-spin state Co³⁺ on the OER activity of CoOOH based electro-catalysts so far.

In this work, we successfully synthesize CoOOH material with high-spin state Co³⁺ through a deliberately designed process involving sulfurization and electro-oxidation route[1,22]. The high-spin state Co³⁺ configuration is verified via superconducting quantum interference device (SQUID), electron paramagnetic resonance (EPR), and X-ray absorption spectroscopy (XAS), exhibiting ferromagnetism behavior at 300 K with unpaired electrons and 3d and 4p orbitals splitting. Density functional theory (DFT) calculation reveals that the appearance of the high-spin state Co³⁺ with magnetic property could be attributed to the emergence of coordinatively unsaturated Co and O atoms at the edges of CoOOH. Projected density of states (PDOS) analysis together with pulse-voltammetry (P-V) measurement indicates that the electronic states around the Fermi level would be significantly increased upon introducing the high-spin state Co³⁺, which could greatly facilitate the electron transfer from electrocatalysts to external circuit via accelerating deprotonation process. As a result, the CoOOH with high-spin state Co³⁺ exhibits splendid OER activity, with an overpotential of 226 mV at a current density of 10 mA cm⁻², outperforming the low-spin state CoOOH by a considerable margin of 148 mV. Moreover, both Co K-edge XAS measurement and

electrochemical analysis confirm the remarkable stability of the CoOOH with high-spin state Co³⁺, which shows negligible structural and activity changes after 200 h Chronopotentiometry (CP) test at 10 mA cm⁻². Therefore, our work provides a comprehensive understanding of the dependency between the spin state of Co³⁺ and corresponding electron transfer kinetics in CoOOH. This knowledge-driven design will greatly benefit the development of outstanding Co-based OER electrocatalysts.

## Results

### Electronic configuration of low-spin and high-spin state Co³⁺ in CoOOH

According to the ligand field theory, electron configuration of Co 3d orbitals could be classified into two primary types[23]. The first type is face-to-face $t_{2g}^*$ orbitals ($d_{xy}$, $d_{xz}$, $d_{yz}$), which are located between CoO₆ octahedral interstices (the left of Fig. 1a). The second one comprises of apex-to-apex $e_g^*$ orbitals ($d_{x^2-y^2}$, $d_{z^2}$), which extend along axis and tend to form bonds between vertices of CoO₆ octahedron (the right of Fig. 1a)[24]. As shown in the energy band (the middle of Fig. 1a), the orbitals closest to the Fermi level are the $t_{2g}^*$ orbitals in the low-spin state Co³⁺, whereas they are the $e_g^*$ orbitals in the high-spin state Co³⁺[17]. A previous report shows that, the electron transfer in apex-to-apex $e_g^*$ orbitals is faster than in face-to-face $t_{2g}^*$ orbitals[25], leading to an

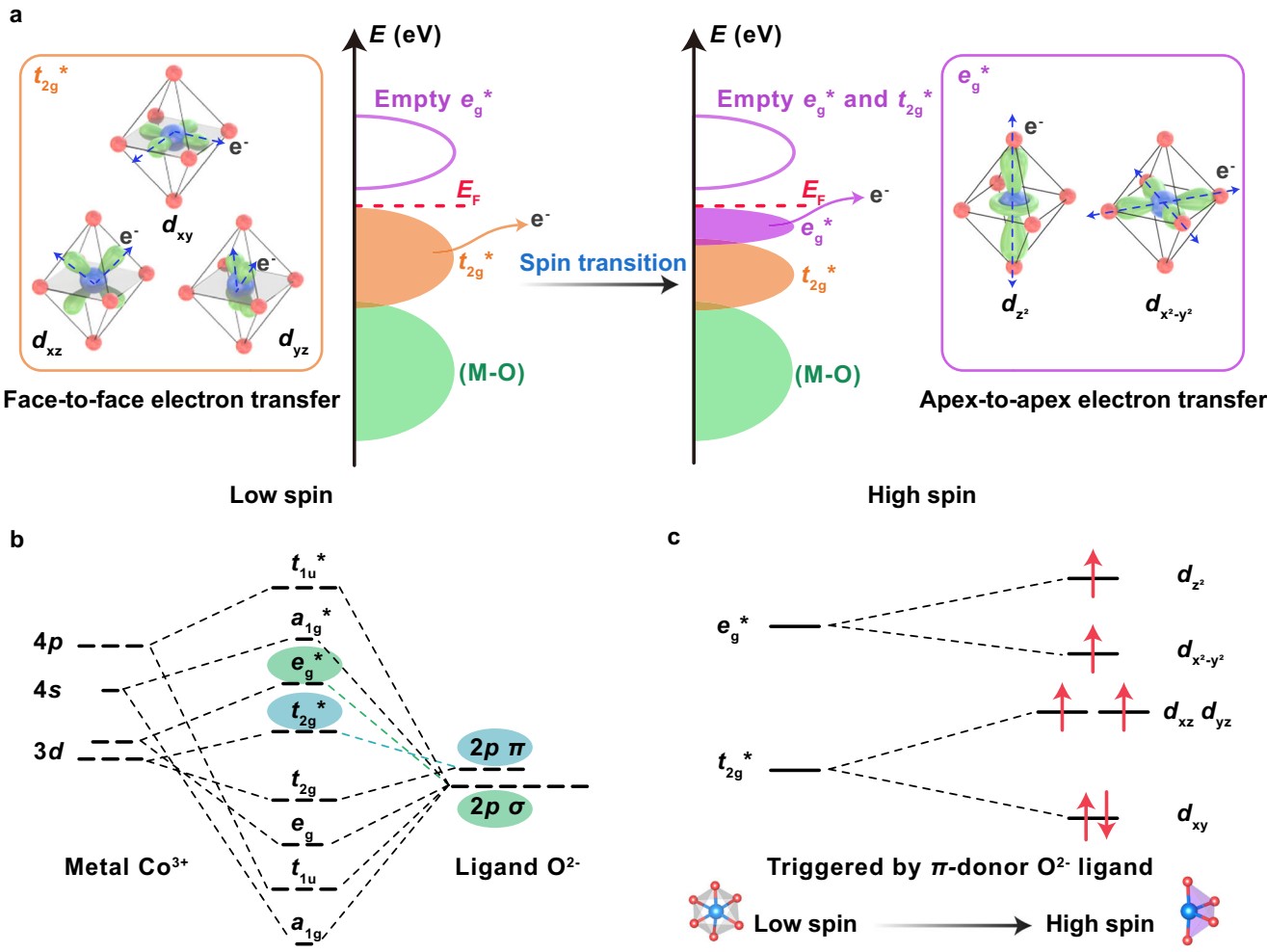

**Fig. 1 | Electronic configuration of low-spin and high-spin state Co³⁺ in CoOOH models. a** Schematic geometry configuration of 3d orbitals ($t_{2g}^*/e_g^*$) in CoOOH, where $t_{2g}^*$ orbitals lie within interstices of the octahedron (left) and $e_g^*$ orbitals extend along axis and tend to form bonds between vertices of the octahedron

(right), and schematic energy band of CoOOH with low-spin and high-spin state Co³⁺ (middle). **b** The molecular orbital diagram of octahedral CoO₆. **c** Electron configuration of 3d electrons in high-spin state CoOOH, triggered by introducing π-donor oxygen ligands via coordinatively unsaturated Co atoms.

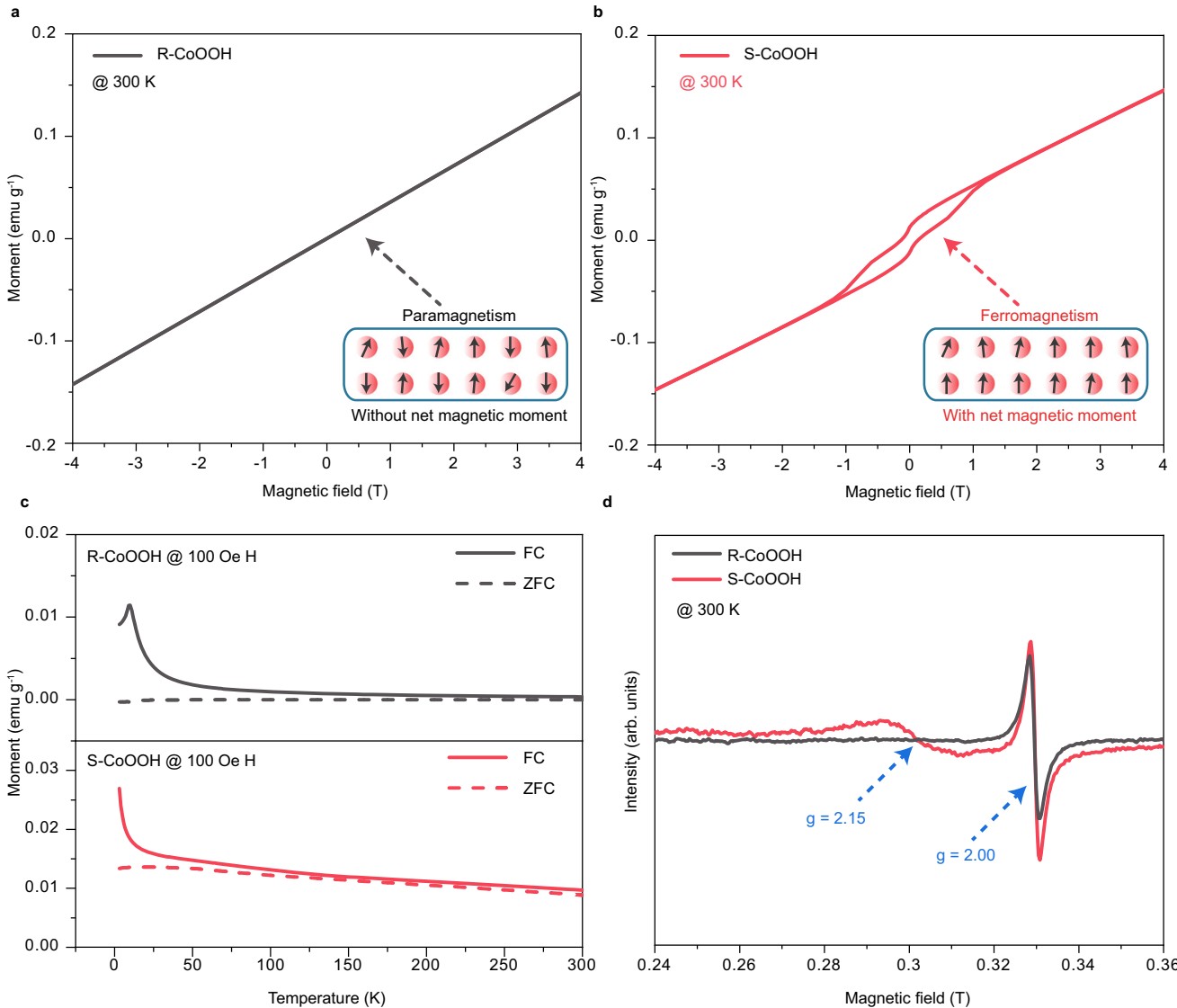

**Fig. 2 | Magnetic analysis of R-CoOOH and S-CoOOH. a, b** Magnetic hysteresis loop of R-CoOOH (a) and S-CoOOH (b) at 300 K, where the insets are magnetic ordering patterns of $Co^{3+}$ ions in low-spin and high-spin states. **c** ZFC (zero field cooled) and FC (field cooled) magnetization curves of R-CoOOH and S-CoOOH as a function of temperature with applied magnetic field **H** = 100 Oe. **d** EPR spectra of R-CoOOH and S-CoOOH recorded at 300 K.

enhanced OER activity. However, there has been no experimental evidence showing the effect of high-spin state $Co^{3+}$ on the OER activity of CoOOH based electrocatalysts so far.

Theoretically, the high-spin state $Co^{3+}$ in CoOOH could be achieved by introducing $\pi$-donor oxygen ligands via coordinatively unsaturated Co atoms[26], which could increase octahedron field splitting energy and electron pairing energy (Fig. 1b, c)[27]. In our previous work, we have successfully prepared nickel hydroxide nanoribbons with unsaturated four-coordinated nickel atoms, via electro-oxidation of nickel sulfides[1,22]. This approach holds promise in providing a guideline for the potential realization of coordinatively unsaturated $Co^{3+}$ ions in CoOOH, which can be advantageous for triggering the spin transition to a high-spin configuration. Motivated by this, a similar approach was employed to prepare CoOOH samples (namely S-CoOOH) through electrochemical oxidation of cobalt sulfides. In contrast, the CoOOH prepared by hydrothermal and electrochemical oxidation processes was used as a benchmark sample, labeled as R-CoOOH. (Detailed information on the preparation methods is provided in the Methods section.)

## Characterization of high-spin state $Co^{3+}$ in S-CoOOH

The synthesized S-CoOOH is firstly analyzed using X-ray diffraction (XRD), Raman spectroscopy, XAS, X-ray photoelectron spectroscopy (XPS), energy dispersive X-ray spectroscopy (EDS) in both high-angle annular dark field scanning transmission electron microscopy (HAADF-STEM) and scanning electron microscopy (SEM), and inductively coupled plasma (ICP) measurements, showing the complete reconstruction of $Co_3S_4$ pre-catalysts to form CoOOH species with negligible residual of S atoms. (Detailed discussions are shown in Supplementary Information, Supplementary Fig. S1-5 and Supplementary Table S1.) Then, the spin configuration in S-CoOOH is investigated. The electronic configuration of 3d orbitals in the high-spin state indicates that when the high-spin state $Co^{3+}$ appears, unpaired electrons would emerge (Fig. 1c). Such an electronic configuration would result in the magnetic property of CoOOH[17,18]. As depicted in Fig. 2a, the magnetic hysteresis (**M-H**) loop of R-CoOOH presents a linear shape without any remanent magnetization, suggesting paramagnetic behavior[28]. Conversely, S-CoOOH reflects a hysteresis loop that reaches saturation, leaving a remanent magnetization when the

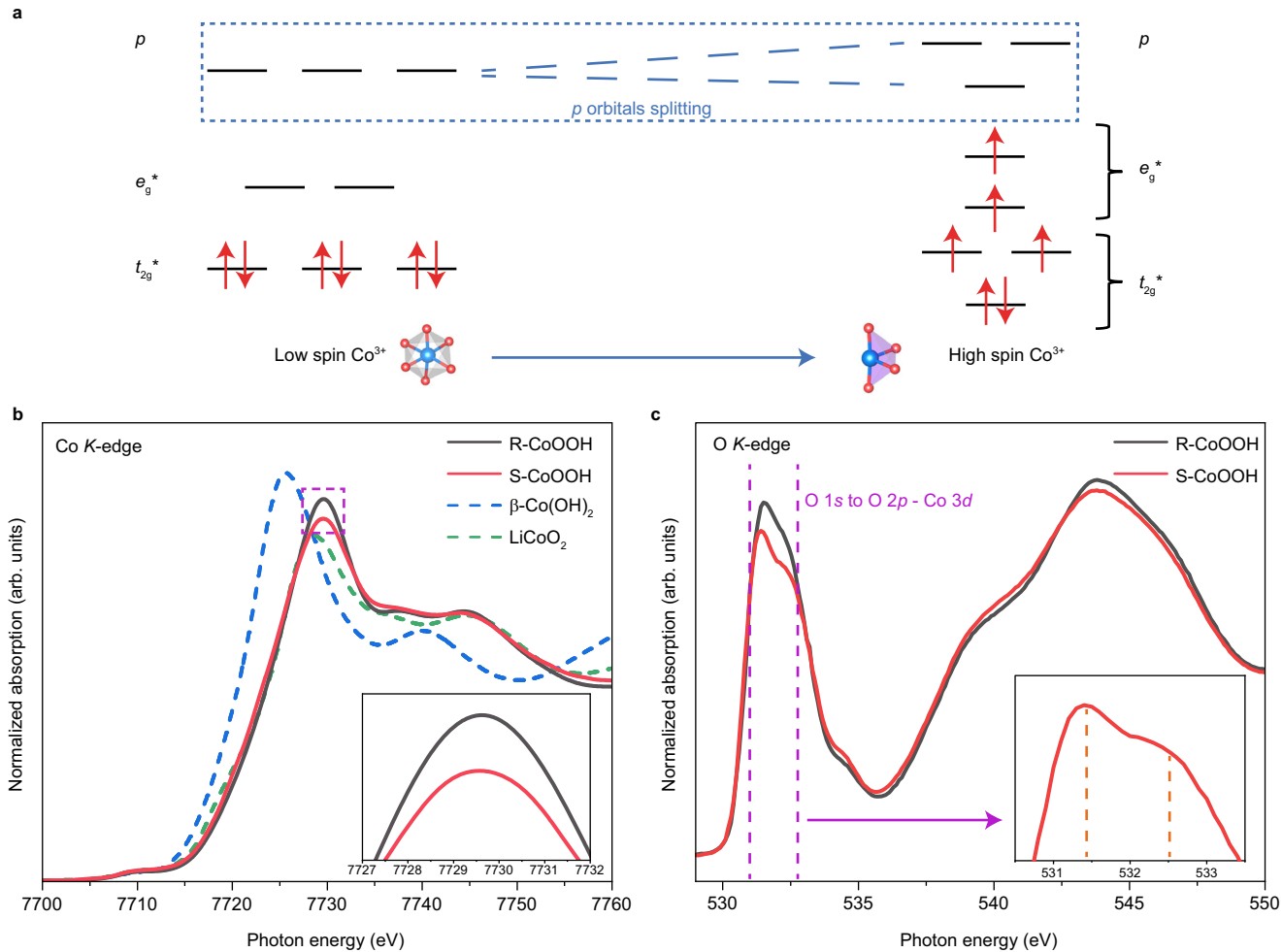

**Fig. 3 | XAS measurements on geometric structure of R-CoOOH and S-CoOOH.** **a** 3d and 4p orbitals configuration of low-spin state $Co^{3+}$ in R-CoOOH and high-spin state $Co^{3+}$ in S-CoOOH. **b** Normalized Co K-edge XAS spectra (the inset shows the enlarged result within white line around 7727 to 7732 eV). **c** Normalized O K-edge XAS spectra (the inset shows the enlarged result of S-CoOOH within pre-edge around 531 to 533 eV).

external magnetic field is reduced to zero (Fig. 2b). These features signify that S-CoOOH exhibits ferromagnetic behavior at 300 K with unpaired electrons. Further analysis of the temperature-dependent magnetization (**M**-T) curves for R-CoOOH (Fig. 2c) reveals an inflection point around 10 K, referred to as the Néel temperature, suggesting a transition from a paramagnetic to an antiferromagnetic state. Considering that 3 K is below the Néel temperature, it confirms antiferromagnetic behavior of R-CoOOH at 3 K (Supplementary Fig. S6), where all electrons are paired up and adjacent valence electrons have opposite spin directions. The non-linear antiferromagnetic character in the **M-H** loop of R-CoOOH may be due to weak antiferromagnetic coupling near the Néel temperature[29,30] or a spin-flop transition influenced by weak magnetic anisotropy[31]. The paramagnetic behavior detected in R-CoOOH at 300 K could be attributed to a collision caused by thermal agitation[32]. Moreover, the slight divergences between ZFC (zero field cooled) and FC (field cooled) curves of both R-CoOOH and S-CoOOH are discussed in Supplementary Fig. S7. Additionally, the effective magnetic moments ($\mu_{eff}$) are calculated using the **M**-T measurements following a Curie-Weiss Law[16]. For R-CoOOH, the calculated $\mu_{eff}$ is 0.09 $\mu_B$ (Supplementary Fig. S8), which is close to those values reported for low-spin state $Co^{3+}$[17]. Similarly, the $\mu_{eff}$ for S-CoOOH calculated to be 0.76 $\mu_B$, suggests the presence of approximately 15 % high-spin state $Co^{3+}$ within the CoOOH structure. This small proportion accounts for the observed low magnetic

properties in S-CoOOH(A detailed discussion is provided in Supplementary Fig. S9).

This could be further verified via EPR spectra (Fig. 2d and Supplementary Fig. S10). As shown in Fig. 2d, a peak at g ~ 2.15 is observed for S-CoOOH, which corresponds to unpaired electrons in high-spin state $Co^{3+}$ ions ($S = 2$)[32]. In contrast, no such peak is evident for R-CoOOH. These results clearly signify that high-spin state $Co^{3+}$ appears in S-CoOOH. Meanwhile, the peaks at g value of ~ 2.00 are observed for both S-CoOOH and R-CoOOH, which could be assigned to oxygen vacancies[33]. It is worth noting that S-CoOOH demonstrates a higher concentration of oxygen vacancies compared to R-CoOOH, which is induced by the coordinatively unsaturated edge $Co^{3+}$[22]. These oxygen vacancies resulting from bond breakage, would concurrently bring coordinatively unsaturated Co and O atoms. The coordinatively unsaturated π-donor ligands could increase the 3d splitting energy and the electron pairing energy, leading to the formation of high-spin state $Co^{3+}$, accompanied by unpaired electrons.

Then, the XAS measurements are utilized to conduct a more in-depth investigation into the presence of high-spin state $Co^{3+}$ in S-CoOOH. Theoretically, as compared with the low-spin state $Co^{3+}$, the high-spin state $Co^{3+}$ would exhibit nondegenerate 3d and 4p orbitals (Fig. 3a)[19]. The Co K-edge XAS spectra reveal that the white line of S-CoOOH becomes broader with lower intensity compared to that of R-CoOOH (the inset of Fig. 3b). It should be noted that the rising edge

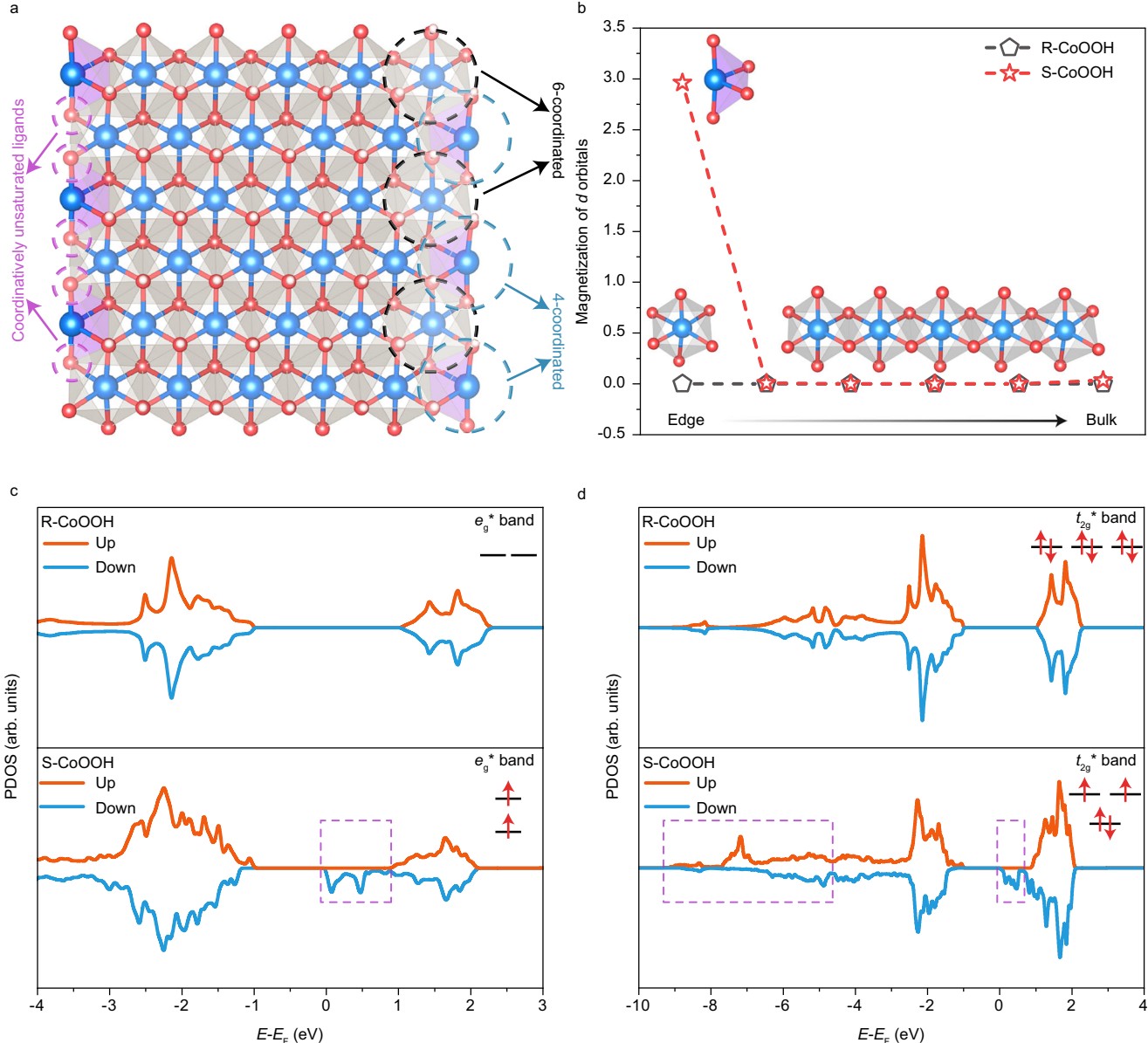

**Fig. 4 | Magnetism distribution of R-CoOOH and S-CoOOH. a** Schematic of optimized model of S-CoOOH structure. **b** The distribution of magnetization obtained from DFT, where the inset is *d*-electron configuration of cobalt cations in different spin states at the edge and in the bulk. **c** The PDOS of cobalt $e_g^*$ band in R-CoOOH and S-CoOOH. **d** The PDOS of cobalt $t_{2g}^*$ band in R-CoOOH and S-CoOOH.

and the white line are generally related to the electron jumping from 1*s* to 4*p* orbitals. A more splitting of 4*p* orbitals is typically represented by a broader white line with lower intensity[6]. Hence, this result suggests that the 4*p* orbitals become nondegenerate in S-CoOOH. Meanwhile, the spin configuration is also analyzed by the pre-edge peak around 7710 eV, corresponding to the Co 1*s*–3*d* transition. And the intensity of the pre-edge peak is correlated to the centrosymmetry of the octahedron in CoOOH[18]. As shown in Supplementary Fig. S11, the pre-edge peak intensity of S-CoOOH is higher than that of R-CoOOH, showing a lower degree of centrosymmetry, which further confirms the presence of high-spin state Co³⁺. The average Co-O bond length is analyzed via fitting the normalized Co *K*-edge Fourier transformed extended X-ray near fine structure (FT-EXAFS) spectra (Supplementary Fig. S12 and Supplementary Table S2). The results show that the fitted Co-O bond length is 1.910 Å for S-CoOOH, longer than that of R-CoOOH (1.903 Å). This elongated Co-O bond length further supports the existence of high-spin state Co³⁺ in S-CoOOH. Moreover, O *K*-edge spectra are

utilized to investigate the 3*d* orbitals configuration. As depicted in Fig. 3c, S-CoOOH shows a lower intensity around 531 to 533 eV (O 1*s* to O 2*p*-Co 3*d* hybrid orbitals), accompanied by more pronounced peak splitting (the inset of Fig. 3c), indicating more splitting of 3*d* band. The Co *L*-edge XAS spectra (Supplementary Fig. S13) show that both $L_3$ and $L_2$ peaks of S-CoOOH become broader with lower intensity than those of R-CoOOH, which also suggests the broadening of 3*d* band in S-CoOOH[18]. These observations strongly corroborate the existence of high-spin state Co³⁺ in S-CoOOH.

## Origin of high-spin state Co³⁺ in S-CoOOH
In this section, the origin of high-spin state Co³⁺ in S-CoOOH is discussed. Firstly, the coordination number (CN) of Co-O bond is fitted based on the Co *K*-edge FT-EXAFS spectra for both S-CoOOH and R-CoOOH. As shown in Supplementary Table S2, it is revealed that the CN of Co-O for S-CoOOH is 5.6, which is lower than that in R-CoOOH (CN = 6). This clearly indicates the presence of coordinatively

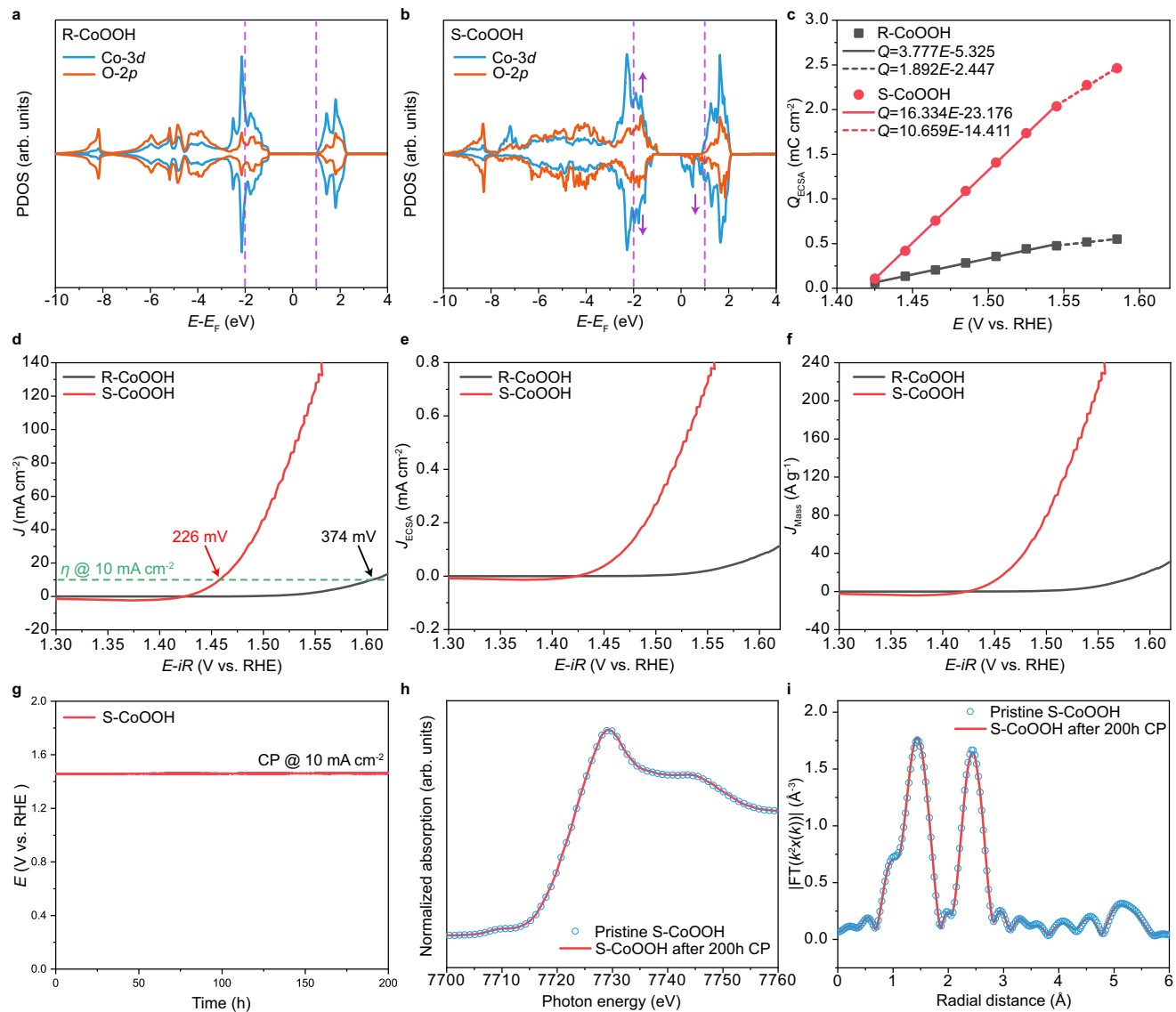

**Fig. 5 | OER activity measurements of R-CoOOH and S-CoOOH in 1 M KOH electrolyte (pH = 13.92). a** PDOS of Co-3$d$ and O-2$p$ orbitals for R-CoOOH. **b** PDOS of Co-3$d$ and O-2$p$ orbitals for S-CoOOH. **c** Charge versus potential from P-V measurements. **d** OER polarization curves after 90% $iR$-correction (R-CoOOH: resistance ≈ 0.77 Ω, overpotential ($\eta$) at 10 mA cm$^{-2}$ ≈ 374 mV. S-CoOOH: resistance ≈ 0.75 Ω, overpotential ($\eta$) at 10 mA cm$^{-2}$ ≈ 226 mV). **e** OER polarization curves with current normalized to ECSA after 90% $iR$-correction, where R-CoOOH is 176.50 cm$^2$

and S-CoOOH is 256.25 cm$^2$. **f** OER polarization curves with current normalized to loading mass after 90% $iR$-correction, where R-CoOOH is 0.64 mg cm$^{-2}$ and S-CoOOH is 0.87 mg cm$^{-2}$. **g** Chronopotentiometry (CP) operation of S-CoOOH at 10 mA cm$^{-2}$ for 200 h. **h** Normalized Co $K$-edge XAS spectra of S-CoOOH before and after CP operation for 200 h. **i** Normalized Co $K$-edge FT-EXAFS spectra of S-CoOOH before and after CP operation for 200 h.

unsaturated Co atoms in S-CoOOH, which would lead to a higher concentration of oxygen vacancies, consistent with the EPR results (Fig. 2d). Additionally, the scanning transmission electron microscopy (STEM) image of S-CoOOH (Supplementary Fig. S14) shows a density of needle-like nano structures. The above experimental result is similar to our previously reported nanoribbon structure Ni(OH)$_2$[22]. As such, DFT calculations are carried out to further investigate the origin of high-spin state Co$^{3+}$ in S-CoOOH, with the CoOOH model in low-spin state as the benchmark[28]. Here, it should be noted that to study the influence of coordinatively unsaturated Co atoms on the spin transition in CoOOH, the S-CoOOH model is constructed similarly to Ni(OH)$_2$ nanoribbon[1,22] and the optimized S-CoOOH structure model is presented in Fig. 4a (The detailed procedure is provided in the Methods section and Supplementary Figs. S15, 16). When the coordinatively unsaturated Co atoms appear, the $\pi$-donor oxygen ligands would be introduced, which would reduce Co-O bond covalency, giving rise to a decrease in Co

valence. As displayed in Supplementary Fig. S17, a small red-shift of 0.47 eV in the photon energy (noted by a purple arrow) is observed in S-CoOOH, indicating a slightly lower Co valence state than that of R-CoOOH. It should be noticed that both Co valence state change induced by hydrogen coverage and spin state could regulate the $e_g^*$ band configuration. Based on S-CoOOH and R-CoOOH models, we find that the hydrogen coverage of S-CoOOH remains consistent with R-CoOOH, preserving a Co:O:H ratio of 1:1:2 (seen in the CONTCAR of optimized models). This excludes the possible contribution from the variation of Co valence induced by hydrogen coverage on the $e_g^*$ band configuration. Hence, the slight decrease of Co valence state is associated with the appearance of high-spin state Co$^{3+}$.

Next, the atomic magnetic moments of coordinatively saturated and unsaturated Co atoms are studied based on DFT simulations (Fig. 4b). For R-CoOOH model, non-magnetic property could be observed for all Co atoms. Similarly, the Co atoms in the bulk of

S-CoOOH with six-coordination display non-magnetism, corresponding to a low-spin configuration of $Co^{3+}$. In contrast, the Co atoms at edge sites of S-CoOOH with four-coordination exhibit ferromagnetism property, with all calculated atomic magnetic moments aligned in the same direction, showing a different spin state configuration. Therefore, the change of magnetism and spin state should be ascribed to the unsaturated coordination at the edge sites in S-CoOOH. Figure 4c reflects $e_g^*$ electronic states of R-CoOOH and S-CoOOH, and the PDOS resulting from unpaired $e_g^*$ electrons in S-CoOOH is highlighted in purple. For R-CoOOH, the spin-up and spin-down densities of $e_g^*$ bands are completely symmetric, which is consistent with previous literature[15]. However, as for S-CoOOH, the $e_g^*$ band becomes broadening compared to R-CoOOH. This result is in line with the $3d$ orbital configuration of high-spin state $Co^{3+}$ (Fig. 3a). Besides, asymmetry is also present in the PDOS for Co $t_{2g}$ band in S-CoOOH (highlighted in purple in Fig. 4d). The loss of spin-related symmetry would bring about ferromagnetism for S-CoOOH, which aligns with those magnetic experimental results. It should be noted that, in view of the calculations for R-CoOOH, the bulk CoOOH displays non-magnetic property, while the surface Co atoms in real structure would lean towards an antiferromagnetic alignment, since the energy of an antiferromagnetic surface is lower than that of a non-magnetic one[10].

### Influence of high-spin state $Co^{3+}$ on OER activity

Here, the influence of high-spin state $Co^{3+}$ on OER activity is explored. The PDOS results reveal that the electronic density of states ranging from −2 to 1 eV around the Fermi level for both Co $3d$ and O $2p$ orbitals become significantly increased after introducing high-spin state $Co^{3+}$ in S-CoOOH at coordinatively unsaturated edge sites (Fig. 5a, b, noted by purple arrows). The more electronic states around the Fermi level would greatly facilitate the electron transfer during the OER process[34]. With this regard, the electron transfer rate from electrocatalyst to external circuit is quantified by P-V measurement[35]. (Detailed protocol is provided in the Methods section and Supplementary Fig. S18a.) Fig. 5c illustrates a linear relationship between the total stored charge $Q_{ECSA}$ and the applied potential, with the bend in the curves ascribed to the varying response of the Tafel slope (Supplementary Fig. S18b). The fitted slope values could represent the charge transfer ability within the corresponding applied potential regions. It could be observed that the obtained slope values of S-CoOOH are all much higher than those of R-CoOOH. These results clearly indicate that high-spin state $Co^{3+}$ electronic configuration could significantly increase the electronic states around the Fermi level, leading to faster electron transfer ability from electrocatalyst to external circuit (Supplementary Fig. S18c, d)[6].

Then, the OER activities of S-CoOOH and R-CoOOH are rigorously compared. Here it should be noted that all electrochemical experiments were performed in Fe-removed 1 M KOH electrolyte. (Detailed information is provided in the Methods section.) 90% $iR$-correction for the LSV polarization curves is utilized to exclude the effect of electrolyte resistance on OER activity. The solution resistance ($R_u$) exhibits a similar value for both S-CoOOH and R-CoOOH, measuring at 0.75 Ω and 0.77 Ω respectively (Supplementary Fig. S19). Figure 5d shows that the 90% $iR$-corrected overpotential of S-CoOOH at 10 mA cm$^{-2}$ is 226 mV, which stands as one of the best reported OER performances among Co-based catalysts (Supplementary Fig. S20 and Supplementary Table S3). Notably, this value displays a significant decrease of 148 mV compared to that of R-CoOOH. Moreover, the corresponding Tafel slope values are 28 mV dec$^{-1}$ for S-CoOOH and 77 mV dec$^{-1}$ for R-CoOOH (Supplementary Fig. S18b). Subsequently, the intrinsic OER activities are investigated by normalizing the current to electrochemical surface area (ECSA) or loading mass. The resulting ECSA values for S-CoOOH and R-CoOOH are 256.25 cm$^2$ and 176.50 cm$^2$, respectively (Supplementary Fig. S21). As shown in Fig. 5e, the intrinsic

overpotential of S-CoOOH normalized to electrochemical surface area (ECSA), is much lower than that of R-CoOOH (Fig. 5e). Consistent results are also evidenced in the results in terms of current density normalized to loading mass (Fig. 5f). The raw electrochemical data without $iR$ correction, including detailed enlargements of two recorded redox peaks corresponding to $Co^{2+/2.5+}$ and $Co^{2.5+/3+}$ redox couples[36], are displayed in Supplementary Fig. S22–24. Moreover, we further conducted electrochemical measurements at high current density. The OER performance of S-CoOOH remains much better than R-CoOOH, indicating that the high-spin state $Co^{3+}$ also plays an important role at high current density (Supplementary Fig. S25). It is noticed that the number of coordinatively unsaturated Co atoms in the high-spin state is very small, as compared with the number of coordinatively saturated Co atoms (Fig. 4a). Yet, the small number of coordinatively unsaturated Co atoms in the high-spin state makes the catalytic property of CoOOH much higher than that of R-CoOOH, demonstrating the superior activity of the high-spin state edge $Co^{3+}$ atoms. Then, the structural and catalytic stability of S-CoOOH is discussed. The CP measurement at 10 mA cm$^{-2}$ reveals that S-CoOOH has high catalytic stability, with negligible decay over approximately 200 h (Fig. 5g and Supplementary Fig. S26). Furthermore, there is no noticeable change observed in the morphology of S-CoOOH before and after the stability test (Supplementary Fig. S27). Additionally, both Co $K$-edge XAS and FT-EXAFS spectra remain nearly identical for pristine S-CoOOH and that subjected to the 200 h CP test (Fig. 5h, i). These results provide strong evidence for the high structural stability of S-CoOOH. As such, we believe that the concept of constructing high-spin state $Co^{3+}$ in CoOOH and its subsequent enhanced electron transfer ability holds promising prospects for the broader application of other oxide-based electrocatalysts.

## Discussion

In summary, we successfully prepare S-CoOOH with high-spin state $Co^{3+}$ by introducing unsaturated coordination at the edge sites. Such a high-spin state $Co^{3+}$ electronic configuration could result in more electronic states around the Fermi level. With this regard, it is found that the electron transfer in apex-to-apex $e_g^*$ orbital for high-spin state CoOOH is much faster in contrast to that in face-to-face $t_{2g}$ orbital for low-spin state CoOOH, hence could significantly improve the intrinsic OER performance. As a result, S-CoOOH exhibits an overpotential of 226 mV at 10 mV cm$^{-2}$, which is 148 mV lower than that of R-CoOOH. Also, S-CoOOH shows long-term structural and catalytic stability in alkaline electrolyte. Our work emphasizes the key role of high-spin state $Co^{3+}$ in CoOOH for superior OER activity.

## Methods

### Materials and reagents

Unless specifically stated otherwise, all materials were purchased from Sigma-Aldrich Co., Ltd. and used as received without further purification. Cobalt sulfate heptahydrate ($CoSO_4·7H_2O$, CAS 10026-24-1, Purity 99 %), cobalt nitrate hexahydrate ($Co(NO_3)_2·6H_2O$, CAS 10026-22-9, 98 %), cobalt hydroxide ($Co(OH)_2$, CAS 21041-93-0, 95 %), lithium cobalt oxide ($LiCoO_2$, CAS 12190-79-3, 99.8 %), boric acid ($H_3BO_3$, CAS 10043-35-3, 99.5 %), hexamethylenetetramine (HMT, $(CH_2)_6N_4$, CAS 100-97-0, 99.5 %), sulfur (S, CAS 7704-34-9, 99.9 %), potassium hydroxide solution (KOH, CAS 1310-58-3, 45 wt % in $H_2O$). Carbon cloth was purchased from CeTech Co., Ltd.

### Synthesis of S-CoOOH

The S-CoOOH sample was synthesized through electro-oxidation of cobalt sulfides. The specific preparation processes in detail are shown as following[22]. The first step was the electrodeposition of metal Co. Typically, carbon cloth (CC, 1 cm×2 cm) was pretreated at 500 °C for 1 hour to ensure a clean and hydrophilic surface. Then, metallic Co was electrodeposited based on the cleaned CC in a two-

electrodes configuration, with CC as the working electrode and Pt as the counter electrode. The electrolyte contained 0.15 M $CoSO_4 \cdot 7H_2O$ and 0.6 M $H_3BO_3$. A constant current density of 10 mA cm$^{-2}$ was applied for 1 hour. The loaded electrode was then rinsed with deionized (DI) water and dried at 60 °C in air for 4 hours. The next step was the formation of $Co_3S_4$. The obtained sample was then heated at 400 °C in an $N_2$ atmosphere for 1 hour. After cooling down to room temperature, it was washed with DI water and then dried at 60 °C in air for 4 hours. Finally, S-CoOOH nanoparticles were obtained through an electrochemical oxidation process, in a three-electrode configuration with Pt as the counter electrode, where -1.58 V versus Hg/HgO (without *iR*-compensation) was applied for 5 hours in 1 M KOH electrolyte.

## Synthesis of R-CoOOH

The reference sample β-CoOOH was prepared through topotactic conversion of β-Co(OH)$_2$[37]. The cleaned CC (1 cm×2 cm) was immersed into 30 mL aqueous solution, containing 5 mmol Co(NO$_3$)$_2$·6H$_2$O and 10 mmol HMT under an intensity ultrasonic treatment[36]. Then the suspensions were transferred to a Teflon-lined stainless steel autoclave (50 mL) and kept at 120 °C for 6 h. The fabricated sample β-Co(OH)$_2$ was washed with DI water six times, followed by drying at 60 °C in air for 4 hours. The electro-oxidation process from Co$^{2+}$ to Co$^{3+}$ (CoOOH) is the same as described before.

## Removal of Fe impurity

1 M KOH electrolyte was purified to remove Fe before utilization. Initially, the KOH solution underwent pre-purging with Grade 4H$_2$ for at least 10 hours[22]. Subsequently, 0.5 g Co(NO$_3$)$_3$·6H$_2$O was added to 30 mL 1 M KOH solution, resulting in the precipitation of Co(OH)$_2$. After centrifugation, Co(OH)$_2$ powder was obtained and then introduced into 50 mL KOH solution accompanied by mechanical agitation. After standing still for 24 hours, the suspension was centrifuged, and the supernatant containing 1 M KOH was decanted into a clean electrochemical cell for further use[6]. The pH value of the purified 1 M KOH was detected to be 13.92.

## Structural characterizations

The XRD patterns were characterized using an X-ray diffraction crystallography equipment (D8, Bruker, Germany) under Cu Kα X-ray irradiation with $\lambda$ = 1.5406 Å. The Raman spectra were recorded by a Raman spectrophotometer (Labram Soleil, Horiab, France) with an excitation wavelength of 514.4 nm. The XPS was measured by a UHV spectrometer (Kratos Axis Ultra DLD, Kratos Analytical, Japan) equipped with an Al Kα X-ray irradiation source (1486.6 eV). The ICP element analysis was obtained through an inductively coupled plasma-optical emission spectrometer (ICP-OES, Avio 500, Perkin Elmer, America). The morphologies and EDS were observed by using filed-emission scanning electron microscopy (Supra 40, Carl Zeiss, Germany), high-resolution transmission electron microscopy and scanning transmission electron microscopy (JEM-2100F, JEOL, Japan).

## XAS characterizations

Cobalt $K$-edge XAS spectra were collected at the XAFCA[38] beamline of Singapore Synchrotron Light Sources (SSLS) using the transmission model, where the storage ring was running at 0.7 GeV with an electron current of approximately 200 mA under top-up mode. Energy calibrations were performed by using standard Cobalt foil. The Cobalt $L$-edge and Oxygen $K$-edge XAS spectra were recorded at the SUV beamline of SSLS. Data acquisition was carried out in total electron yield mode with a photon energy resolution of 350 meV. The photon energy was calibrated by referencing the characteristic intensity dip, which was associated with the carbon contamination of the beamline optical components at 284.4 eV. All XAS spectra were normalized to the incident photon intensity ($I_O$) monitored by the focusing mirror.

Athena was used for energy calibration, background removal, and Fourier Transform (FT)[39].
  Fitting parameters:
  Method: least-squares
  Space: R
  K range: 3-13 Å
  R range: 1-2.85 Å
  Software: Artemis
  Initial structure: standard β-CoOOH
  Theoretical scattering path calculation code: FEFF6
  K weight during EXAFS fitting: 1,2,3
  K weight used in plot: 3

## Magnetic measurements

The magnetic properties of the samples were studied by analyzing EPR and SQUID data. The EPR spectra were conducted by an EPR spectrometer (FA200, JEOL, Japan) at an X-band frequency (around 9.20 GHz). Moreover, the **M**-**H** and **M**-T plots were obtained through a SQUID magnetometer (MPMS3, Quantum Design, America).

## Effective magnetic moment calculations

The effective magnetic moments ($\mu_{eff}$) of cobalt ions were investigated using **M**-T measurements following a Curie-Weiss Law[16,17]. Here, the **M**-T measurements were conducted with a magnetic field of **H** = 100 Oe under field cooling procedures for both R-CoOOH and S-CoOOH. The susceptibilities derived from the magnetizations ($\chi = M/H$) obey a Curie-Weiss law: $\chi = C/(T - T_c)$ where $C$ is Curie constant, and $T_c$ is Curie-Weiss temperature. The Curie constant ($C$) was extracted from the inverse susceptibility ($\chi^{-1}$)-temperature ($T$) plot, which was fitted based on the **M**-T measurements data. Then, the $\mu_{eff}$ for both R-CoOOH and S-CoOOH were calculated through the equation $\mu_{eff} = \sqrt{8C}\mu_B$, where $\mu_B$ is Bohr magneton.

## Electrocatalytic OER measurements

Electrochemical measurements were performed using an electrochemical workstation (VPM3, Bio-logic Inc, France) in a typical three-electrode setup with 30 mL 1 M aqueous KOH as the electrolyte, as-prepared sample based on carbon cloth as the working electrode, Pt as the counter electrode, and Hg/HgO as the reference electrode. The as-measured potentials (versus Hg/HgO) were calibrated with respect to the reversible hydrogen electrode (RHE). Some detailed experimental information is given below.

## Linear sweep voltammetry (LSV) measurements

To get a precise overpotential, the scan rate was set at a slow value of 0.1 mV s$^{-1}$ to minimize capacitive currents. The conversion between the potentials versus Hg/HgO and RHE was calculated by using Nernst Eq. (1) as below[8].

$$E_{RHE} = E_{Hg/HgO} + 0.098 + (0.059 \times pH)\, V \qquad (1)$$

## Tafel plots

These polarization curves were plotted as the overpotential versus the log current, derived from LSV measurements. The Tafel slope was obtained by fitting the linear portion of the Tafel plots according to Tafel Eq. (2)[5], where η is overpotential in V, $J$ is the current density in mA cm$^{-2}$, and $b$ is Tafel slope in mV dec$^{-1}$.

$$\eta = b\log[J] + a \qquad (2)$$

## *iR* compensation

The solution resistance ($R_u$) was obtained by impedance spectroscopy. This measurement was conducted at open circuit potential with the

frequency set from 10 mHz to 100 kHz. From the diameter of the semicircle in the Nyquist plots, $R_u$ was approximated. The potential was corrected by 90 % of $R_u$, according to Eq. (3)[22].

$$E_{iR\text{corrected}} = E - iR_u \qquad (3)$$

## Electrochemical surface area (ECSA)

The ECSA could be estimated by measuring the non-Faradaic capacitive current associated with double-layer charging from the scan-rate dependence of cyclic voltammetry (CV), according to Eq. (4) below[40].

$$ECSA = \frac{C_{DL}}{C_s} \qquad (4)$$

Where $C_{DL}$ is the double-layer capacitance, and $C_s$ is the specific capacitance of any investigated electrode material. As reported, here we used general specific capacities of $C_s = 0.040$ mF cm$^{-2}$ in 1 M KOH for estimating our surface area. $C_{DL}$ could be extracted from recording CVs at various scan rates. In specific, the potential range of the CV measurements was selected between 0.60 and 0.70 V (versus RHE), in order to avoid the Faradaic and redox processes taking place. Also, the scan rates varied from 0.01, 0.02, 0.03, 0.04 to 0.05 mV s$^{-1}$. These tests were all conducted in a static electrolyte. Here, the slope of the resulting charging current vs. scan rate plot was approximately regarded as $C_{DL}$.

## Pulse-voltammetry (PV) Tests

This electrochemical setup was performed in a glass cell at room temperature[35]. Before the OER scan, an activation step under 10 mA cm$^{-2}$ for 1 h in 1 M KOH electrolyte was applied for R-CoOOH and S-CoOOH samples to leach out Co$^{2+}$. The potential was kept at a lower potential ($E_1 = 1.40$ V) for 300 s, and kept at a higher potential ($E_h$) for 6 s before returning to $E_1$ for 6 s. This cycle was repeated while increasing from 1.42 V to 1.58 V in 20 mV per step with $E_1$ unchanged. The transferred charge normalized to ECSA during each cycle was evaluated by integrating the current pulse per ECSA over time.

## Theoretical calculations

DFT calculations were performed by the Vienna Ab initio Simulation Package (VASP) combined with Perdew–Burke–Ernzerhof (PBE) functional[41] and Projector augmented-wave (PAW) pseudopotentials[42]. DFT + U (U-J = 3.52 for Co)[43] and DFT-D3 dispersion correction[44] were considered in all calculations. Plane wave kinetic energy cutoff was set to 500 eV and the energy and force convergence criteria were set to 10$^{-5}$ eV and 0.03 eV Å$^{-1}$, respectively. The gamma centered k-points mesh of $14 \times 14 \times 14$ and $4 \times 8 \times 1$ was used for densities of states (DOS) calculations of β-CoOOH primitive cell and the CoOOH edge structure, respectively. The electronic structure analysis was performed by VASPKIT[45] and the atomic structures were visualized by VESTA[46].

## Data availability

All the data supporting the findings of this study are included within the paper and its supporting files and are available from the corresponding authors on request.

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

## Acknowledgements

This research is supported by the National Research Foundation, Singapore, under its Competitive Research Programme (Award No.: NRF-CRP26-2021-0003), Ministry of Education, Singapore, under its Academic Research Fund (AcRF) Tier 2 (Award No.: MOE-T2EP501220010).

## Author contributions

X.Z. and H.Y.Z. contributed equally to this work. X.Z. and X.P.W. conceived the idea. X.Z. completed the original draft writing. H.Y.Z. and X.P.W. contributed to the writing-review and editing. X.Z. and H.Y.Z. performed synthesis and electrochemical measurements of the samples. J.C.Y., Y.F.M., H.A., and H.W. were responsible for the analysis of electrochemical results. Q.H.Z. and J.S.C. performed magnetic tests and analysis. Q.Z. and Z.G.Y. carried out DFT simulations. S.B.X., C.Z.D., and C.W. were responsible for the XAS characterizations. Y.M.Z. conducted the STEM and HRTEM tests. J.M.X. oversaw the overall project and preparation of the manuscript.

## Competing interests

The authors declare no competing interests.
