## [Peer Review File · Nature Communications]

REVIEWER COMMENTS

Reviewer #1 (Remarks to the Author):

Zhang et al. reported the synthesis of CoOOH with high-spin state Co³⁺ by introducing coordinatively unsaturated Co atoms for OER in this work. They found that the high-spin state CoOOH showed higher OER activity than the low-spin state CoOOH because of faster electron transfer in the apex-to-apex *eg**orbitals. Although the high-spin state Co³⁺ has been theoretically reported for improved OER, the experimental confirmation is still missing. This work may provide a direct evidence to the long-sought origin.

1. As R-CoOOH and S-CoOOH were fabricated by different methods, how to make sure the spin played the dominant role for the improved activity? Other factors, such as defects, may contribute to the phenomenon. The EPR clearly showed that the density of oxygen vacancies in S-CoOOH was higher than that in R-CoOOH.
2. As S-CoOOH was obtained by oxidizing Co₃S₄, there should have the heterojunctions and S-incorporation, which may improve the performance. Additionally, Co₃S₄ has higher conductivity than R-CoOOH.
3. The test was carried out at low current density. How about high current density? Did the high-spin play important role at high current density?
4. Is it possible to have a simple way for the investigation easily? For example, using Co₃O₄ as pre-catalyst for CoOOH?

Reviewer #2 (Remarks to the Author):

This manuscript by Zhang et al. reports the synthesis of high-spin Co³⁺ in CoOOH via electro-oxidation of cobalt sulfides (named as S-CoOOH) as efficient electrocatalyst for water oxidation. The authors performed a combination of detailed characterizations on the magnetic properties and the spin state of Co using EPR, SQUID and XAS, which prove the high spin state of Co³⁺ in their samples. The results are quite convincing. The authors also demonstrated, by comparing with normal CoOOH sample with low spin state of Co³⁺ (R-CoOOH), that the S-CoOOH exhibit a much high OER activity than R-CoOOH. Although previous theoretical works predict that the *eg* electrons at the high spin of Co³⁺ lead to high OER activity, this is the first experimental work directly prove the proposed mechanism. Therefore, overall, I think this is a very interesting work and should have broad impact for the field of electrocatalysts. I would suggest to accept if a few the following issues be addressed.

1. The switch of low spin state in R-CoOOH to high spin state in S-CoOOH only changes the spin configuration of Co 3d, but does change the total density state. However, in Figure 1c, the density of state, and especially the total occupied density of state for high spin increases quite a lot. This is not correct. Please revise it accordingly.
2. XPS show there is S left in the S-CoOOH sample. But XPS is a quite surface sensitive technique (~ a few nm). Is it the same for the whole sample?

3. It is not still not clear why the electro-oxidation synthesized S-CoOOH exhibit ferromagnetic properties, while the R-CoOOH not (as the authors claimed there is no S left in S-CoOOH). I would suggest the authors make more comparison on the morphology and the atomic structure of R-CoOOH and S-CoOOH. TEM image and the XAS data might provide some clues for this.
4. The S-CoOOH show very good electrochemical stability. Is there any morphological or structural changes after 200 hours operation?
5. I am wondering whether electron doping in R-CoOOH using tetravalent or pentavalent cations (e.g. Ta) can also stabilize high spin state of Co, because the extra electron will occupy eg state.

Reviewer #3 (Remarks to the Author):

Zhang et al reported that the electrochemical oxidation-derived CoOOH nanoparticles, labeled as S-CoOOH in this manuscript, exhibited a significant enhancement of OER activity, compared to the bulk CoOOH (labeled as R-CoOOH). Based on the results from magnetic analysis, EPR, XAFS, and DFT calculations, the authors attributed this enhancement to the spin state transition of Co ions in S-CoOOH, from low-spin (LS) to high-spin (HS), which was induced by the decrease in the coordination number of Co ions. Although the topic raised in this work, i.e, the role of spin state on the catalysis, is interesting, several issues should be addressed before publication as following:

1. The authors claimed that the CoOOH sample with HS-state Co ions was synthesized for the first time. However, similar result can be found in previous works (for example, *Angew. Chem.* 2015, 127, 8846–8851). Moreover, the existence of HS-state Co ions has been frequently reported in nanostructured cobalt oxides (see *Nat. Commun.* 2016, 7, 11510; *Angew. Chem. Int. Ed.* 2023, e202216837).
2. For nanostructured metal oxides, the decrease in the coordination for metal ions is usually associated with the presence of oxygen vacancies. Consequently, the valence state of metal ions would be reduced. As shown in Fig. 3b, a red shift in the Co K-edge X-ray absorption suggested a lowered Co valence state in S-CoOOH. It should notice that both the Co valence state and spin state can regulate the eg orbital configuration. Therefore, to explore the role of spin state, the author should exclude the contribution from the reduction in the Co valence state.
3. From the XAFS fitting results (Fig. S12 and Table S2), the Co-O coordination number in S-CoOOH was 5.7, just a little lower than that in R-CoOOH (6). This means that the proportion of coordinatively unsaturated Co atoms in S-CoOOH is small. Moreover, the S-CoOOH structure model as shown in Fig. 4a illustrated that the coordinatively unsaturated Co atoms only appeared at edge sites. Those results suggest that even if the coordinatively unsaturated Co atoms were in HS state as the authors stated, the proportion of them would be small in S-CoOOH. The author should clearly illustrate this point in the manuscript.
4. For R-CoOOH, the authors stated that it would be paramagnetic at high temperature (300 K) and antiferromagnetic at low temperature (3 K). Ideally, both the paramagnetic and antiferromagnetic states give rise to a linear dependence of magnetization on the magnetic field. However, in Fig. S9, the magnetic hysteresis loops at 3 K for R-CoOOH exhibited a non-linear behavior. Why? On the other hand,

for S-CoOOH, the magnetic hysteresis loops at 3 K revealed that both the remanent magnetization and the coercive field were reduced to zero. Why? Please explain the above results.

5. In Fig. 2c, the ZFC and FC magnetization curves diverge for both R-CoOOH and S-CoOOH. Why?

6. The magnetic moments and interactions would give evidence of spin configuration, which can be obtained by fitting the temperature-dependent magnetizations via the Curie-Weiss law. This fitting is suggested to be carried out for both the samples.

7. In Co K-edge XAS spectra (Fig. 3b), the pre-edge peak around 7710 eV corresponds to the Co 1s-3d transition, which would evidence the spin configuration of Co ions. A detailed analysis on this peak is suggested. Additionally, Co L-edge XAS is also suggested to be carried out to get more evidence of the spin state transition.

8. The Co-O bond length for HS state should be larger than that for LS state. Please give an analysis on the Co-O bond length from Co k-edge XAFS for both the samples.

9. Dose the calculated PDOS support that the magnetic ground state of R-CoOOH is antiferromagnetic?

10. The CV curves (Fig. S17c) for R-CoOOH are not centered at 0 mA. Why?

11. In Fig. S19a, S-CoOOH exhibited two distinct redox peaks but R-CoOOH did not. Why? If the peak around 1.4 V belongs to the Co^{III/IV} redox process, the OER active sites for S-CoOOH would be Co^{IV} ions. In this case, how does the spin state of Co³⁺ ions influence the OER activity?

12. "CoLiO₂" in Fig. S13b and Table S2 should be denoted as "LiCoO₂".

Response to referees

We thank the three referees for taking the time to carefully review the manuscript and for giving many useful suggestions. The manuscript has been revised according to their comments and the changes have been highlighted in the revised version. We feel the quality of the paper has been improved greatly thanks to the input from the referees. Below is a point-by-point response.

Reviewer #1

Remarks to the Author: Zhang et al. reported the synthesis of CoOOH with high-spin state Co³⁺ by introducing coordinatively unsaturated Co atoms for OER in this work. They found that the high-spin state CoOOH showed higher OER activity than the low-spin state CoOOH because of faster electron transfer in the apex-to-apex *eg** orbitals. Although the high-spin state Co³⁺ has been theoretically reported for improved OER, the experimental confirmation is still missing. This work may provide a direct evidence to the long-sought origin.

1. As R-CoOOH and S-CoOOH were fabricated by different methods, how to make sure the spin played the dominant role for the improved activity? Other factors, such as defects, may contribute to the phenomenon. The EPR clearly showed that the density of oxygen vacancies in S-CoOOH was higher than that in R-CoOOH.

Response: We are very grateful for the referee's comment. We agree with the reviewer's comment that the density of oxygen vacancies in S-CoOOH is higher compared to R-CoOOH. In this work, we find that the higher concentration of oxygen vacancies is induced by the coordinatively unsaturated edge Co³⁺ in S-CoOOH, which is the reason for the formation of high-spin state Co³⁺. Next, we discuss why in our work, these oxygen vacancies resulting from bond breakage, contribute to the emergence of high-spin configuration. Specifically, the breaking of the Co-O bond would concurrently bring coordinatively unsaturated Co and O atoms (Fig.R1-1a). These coordinatively unsaturated π -donor ligands could increase the 3d splitting energy and electron pairing energy, leading to the formation of high-spin state Co³⁺, accompanied by unpaired electrons. This could be identified via magnetic measurements. As shown in

Fig.R1-1b, the magnetism in S-CoOOH is induced by the edge coordinatively unsaturated Co^{3+} with oxygen vacancies, while there is no magnetic moment in the R-CoOOH model.

Fig.R1-1 **a** Schematic of optimized model of S-CoOOH structure. **b** The distribution of magnetization obtained from density functional theory (DFT), where the inset is *d*-electron configuration of cobalt cations in different spin states at the edge and in the bulk.

Besides, other possible factors *i.e.*, heterojunctions and S-incorporation are also considered in our work. Fig.R1-2a shows that there is negligible S for S-CoOOH after electrochemical oxidation. Further, the Co *K*-edge Fourier transformed extended X-ray near fine structure (FT-EXAFS) results indicate that all Co-S bonds have been broken in S-CoOOH. As such, the potential formation of $\text{Co}_3\text{S}_4/\text{CoOOH}$ or S-incorporation could be excluded. Therefore, the high-spin state Co^{3+} induced by the edge unsaturated coordination with oxygen vacancies plays a dominant role in the improved activity.

Fig.R1-2 **a** High-resolution X-ray photoelectron spectroscopy (XPS) spectra of S *2p* peak for pre-catalyst Co_3S_4 and post-activation S-CoOOH. **b** Co *K*-edge FT-EXAFS spectra of pre-catalyst Co_3S_4 and post-activation S-CoOOH.

To address the comment, we have incorporated the proposed modifications into our manuscript (line 144-150, page 6-7, highlighted in yellow): Meanwhile, the peaks at g value of ~ 2.00 are observed for both S-CoOOH and R-CoOOH, which could be assigned to oxygen vacancies³³. It is worth noting that S-CoOOH demonstrates a higher concentration of oxygen vacancies compared to R-CoOOH, which is induced by the coordinatively unsaturated edge Co^{3+} ²². These oxygen vacancies resulting from bond breakage, would concurrently bring coordinatively unsaturated Co and O atoms. The coordinatively unsaturated π -donor ligands could increase the $3d$ splitting energy and electron pairing energy, leading to the formation of high-spin state Co^{3+} , accompanied by unpaired electrons.

In addition, we have made further revisions in another section of our manuscript (line 189-194, page 9-10, highlighted in yellow): Firstly, the coordination number (CN) of Co-O bond is fitted based on the Co K -edge FT-EXAFS for both S-CoOOH and R-CoOOH. As shown in Supplementary Table S2, it is revealed that the CN of Co-O for S-CoOOH is 5.6, which is lower than that in R-CoOOH (CN = 6). This clearly indicates the presence of coordinatively unsaturated Co atoms in S-CoOOH, which would lead to a higher concentration of oxygen vacancies, consistent with the EPR results (Fig. 2d).

2. As S-CoOOH was obtained by oxidizing Co_3S_4 , there should have the heterojunctions and S-incorporation, which may improve the performance. Additionally, Co_3S_4 has higher conductivity than R-CoOOH.

Response: We are very grateful for the referee's comment. In our work, we conduct a long-time electrochemical oxidation procedure on the Co_3S_4 to fully convert it into CoOOH species. This can be supported by the following results: 1) X-ray diffraction (XRD), Raman spectra, and X-ray absorption spectroscopy (XAS) (Fig.R1-3), showing that the bulk Co_3S_4 is completely converted into CoOOH. 2) The Co K -edge FT-EXAFS results indicate that all Co-S bonds have been broken in S-CoOOH after electrochemical oxidation (Fig.R1-3d). 3) Energy Dispersive X-ray Spectroscopy (EDS) in both Scanning Transmission Electron Microscopy (STEM) and Scanning Electron Microscopy (SEM) results (Fig.R1-4, 1-5), indicate an almost negligible signal of S element in S-CoOOH. 4) The XPS analysis (Fig.R1-6) reveals

an absence of S peaks in the S-CoOOH sample, implying no S left on the S-CoOOH surface. 5) In addition, the Inductively Coupled Plasma (ICP) results (Table R1-1) also confirm an almost negligible S signal in S-CoOOH. As such, these combined results indicate a complete conversion of Co_3S_4 into CoOOH, excluding the possible formation of heterojunctions or S-incorporation.

Fig.R1-3 Structural characterization of as-prepared Co_3S_4 and after activation S-CoOOH, where Co_3S_4 is marked in black and S-CoOOH is marked in red. **a** XRD patterns of Co_3S_4 and S-CoOOH with standard peaks of Co_3S_4 and $\beta\text{-CoOOH}$ listed. **b** Raman spectra of Co_3S_4 and S-CoOOH. **c** Normalized Co K-edge XAS spectra of Co_3S_4 and S-CoOOH. **d** FT-EXAFS spectra of Co K-edge of Co_3S_4 and S-CoOOH.

Fig.R1-4. The STEM-EDS results of S-CoOOH. **a** The quantitative elemental analysis. **b** EDS analysis area. **c-e** EDS mapping of oxygen (dark cyan), cobalt (purple), sulfur (yellow). No S signal is detected.

Fig.R1-5 The SEM-EDS results of S-CoOOH. **a** The quantitative elemental analysis. **b** EDS analysis area. **c-e** EDS mapping of oxygen (dark cyan), cobalt (purple), sulfur (yellow). No S signal is detected.

Fig.R1-6 High-resolution XPS spectra of S 2p peak for pre-catalyst Co_3S_4 and post-activation S-CoOOH.

Table R1-1. ICP results of S-CoOOH, showing the corresponding mass/atomic ratios

Element	Co	S
Mass ratio (w/w%)	36.4	0.28
Atomic ratio (at%)	98.6	1.40

To address the comment, we have incorporated the proposed modifications into our manuscript (line 112-118, page 5-6, highlighted in yellow): The synthesized S-CoOOH is firstly analyzed using X-ray diffraction (XRD), Raman spectroscopy, XAS, X-ray photoelectron spectroscopy (XPS), Energy Dispersive X-ray Spectroscopy (EDS) in both High-Angle Annular Dark Field Scanning Transmission Electron Microscopy (HAADF-STEM) and Scanning Electron Microscopy (SEM), and Inductive Coupled Plasma (ICP) measurements, showing the complete reconstruction of Co_3S_4 pre-catalysts to form CoOOH species with negligible residual of S atoms. (Detailed discussions are shown in Supplementary Information, Supplementary Fig. S1-5 and Supplementary Table S1).

3. The test was carried out at low current density. How about high current density? Did the high-spin play important role at high current density?

Response: We sincerely appreciate the referee's insightful suggestions. To investigate the role of high spin

at high current density, we further conduct a linear sweep voltammetry (LSV) test at a high potential range. It is revealed that the current density of S-CoOOH in the high potential region is much higher than that of R-CoOOH (Fig.R1-7a). Moreover, the electrochemical surface area (ECSA)-normalized current density of S-CoOOH is also higher than that of R-CoOOH within the high potential region, showing excellent intrinsic activity (Fig.R1-7b). As such, it could be known that the high-spin state Co^{3+} also plays an important role at high current density.

Fig.R1-7 OER activities of R-CoOOH and S-CoOOH. **a** OER polarization curves. **b** OER polarization curves normalized to ECSA. (Supplementary Fig. S24 in Supplementary Information)

To address the comment, we have incorporated the proposed modifications into our manuscript (line 274-276, page 13, highlighted in yellow): Moreover, we further conducted electrochemical measurements at high current density. The OER performance of S-CoOOH remains much better than R-CoOOH, including that the high spin Co^{3+} also plays an important role at high current density (Supplementary Fig. S24).

Supplementary Fig. S24 OER activities of R-CoOOH and S-CoOOH. **a** OER polarization curves. **b** OER polarization curves normalized to ECSA.

4. Is it possible to have a simple way for the investigation easily? For example, using Co_3O_4 as pre-catalyst for CoOOH ?

Response: We are very grateful for the referee's comment. To investigate the role of spin state on the OER activity, it is required that the pre-catalysts are fully converted into the CoOOH species. In this work, we also prepared the Co_3O_4 samples, as shown in Fig.R1-8. Different from Co_3S_4 samples, after long-time electrochemical oxidation treatment, there is no obvious phase transition. Hence, it is challenging to use Co_3O_4 for the investigation.

Fig.R1-8 XRD patterns of pre-catalyst Co_3O_4 and post-activation sample, with standard peaks of Co_3O_4 listed.

Reviewer #2

Remarks to the Author: This manuscript by Zhang et al. reports the synthesis of high-spin Co^{3+} in CoOOH via electro-oxidation of cobalt sulfides (named as S- CoOOH) as efficient electrocatalyst for water oxidation. The authors performed a combination of detailed characterizations on the magnetic properties and the spin state of Co using EPR, SQUID and XAS, which prove the high spin state of Co^{3+} in their samples. The results are quite convincing. The authors also demonstrated, by comparing with normal CoOOH sample with low spin state of Co^{3+} (R- CoOOH), that the S- CoOOH exhibit a much high OER activity than R- CoOOH . Although previous theoretical works predict that the eg electrons at the high spin of Co^{3+} lead to high OER activity, this is the first experimental work directly prove the proposed mechanism. Therefore, overall, I think this is a very interesting work and should have broad impact for the field of electrocatalysts. I would suggest to accept if a few the following issues be addressed.

1. The switch of low spin state in R- CoOOH to high spin state in S- CoOOH only changes the spin configuration of Co 3d, but does change the total density state. However, in Figure 1c, the density of state, and especially the total occupied density of state for high spin increases quite a lot. This is not correct. Please revise it accordingly.

Response: We appreciate the referee's insightful suggestion. According to the reviewer's advice, we have revised Fig. 1c accordingly. (Fig. R2-1)

Fig.R2-1 Electronic configuration of low-spin and high-spin state Co^{3+} in $CoOOH$ models. **a** Schematic geometry configuration of $3d$ orbitals (t_{2g}^*/e_g^*) in $CoOOH$, where t_{2g}^* orbitals lie within interstices of the octahedron (left) and e_g^* orbitals extend along the axis and tend to form bonds between vertices of the octahedron (right), and schematic energy band of $CoOOH$ with low-spin or high-spin state Co^{3+} (middle). **b** The molecular orbital diagrams of octahedral CoO_6 . **c** Electron configuration of $3d$ electrons in high-spin state $CoOOH$, triggered by introducing π -donor oxygen ligands via coordinatively unsaturated Co atoms. (Fig. 1 in the revised manuscript)

2. XPS show there is S left in the S- $CoOOH$ sample. But XPS is a quite surface sensitive technique (\sim a few nm). Is it the same for the whole sample?

Response: We are very grateful for the referee's comment. We agree with the reviewer's comment that XPS analysis reveals the signal of the surface (a few nm). In the high-resolution XPS spectra of S $2p$ for Co_3S_4 before and after activation (Fig.R2-2a), no peaks assigned to S $2p$ are observed in S- $CoOOH$, indicating that S is totally removed during the OER process. To investigate the bulk information, we have conducted Fourier transformed extended X-ray near fine structure (FT-EXAFS) and Inductively Coupled Plasma (ICP)

characterizations. As shown in the FT-EXAFS spectra, the Co-S bond peak disappears after electrochemical oxidation, indicating that all Co-S bonds are broken (Fig.R2-2b). In addition, the ICP results show an almost negligible signal of S element for the whole sample (Table R2-1). Based on these results, it could be concluded that the whole Co_3S_4 pre-catalyst is completely transformed into CoOOH species after the long-time electrochemical oxidation process, with negligible S left.

Fig.R2-2 a XPS spectra of S 2p peak for Co_3S_4 and S-CoOOH. b Co K-edge FT-EXAFS spectra of Co_3S_4 and S-CoOOH.

Table R2-1. ICP results of S-CoOOH, showing the corresponding mass/atomic ratios

Element	Co	S
Mass ratio (w/w%)	36.4	0.28
Atomic ratio (at%)	98.6	1.40

To address the comment, we have incorporated the proposed modifications into our manuscript (line 112-118, page 5-6, highlighted in yellow): The synthesized S-CoOOH is firstly analyzed using X-ray diffraction (XRD), Raman spectroscopy, XAS, X-ray photoelectron spectroscopy (XPS), Energy Dispersive X-ray Spectroscopy (EDS) in both High-Angle Annular Dark Field Scanning Transmission Electron Microscopy (HAADF-STEM) and Scanning Electron Microscopy (SEM), and Inductive Coupled Plasma (ICP) measurements, showing the complete reconstruction of Co_3S_4 pre-catalysts to form CoOOH species with negligible residual of S atoms. (Detailed discussions are shown in Supplementary Information, Supplementary Fig. S1-5 and Supplementary Table S1).

3. It is not still not clear why the electro-oxidation synthesized S-CoOOH exhibit ferromagnetic properties, while the R-CoOOH not (as the authors claimed there is no S left in S-CoOOH). I would suggest the authors make more comparison on the morphology and the atomic structure of R-CoOOH and S-CoOOH. TEM image and the XAS data might provide some clues for this.

Response: We are very grateful for the referee's comment. In our work, the ferromagnetic properties of S-CoOOH are due to the coordinatively unsaturated edge Co atoms. As shown in Table R2-2, the fitted Co-O coordination number in S-CoOOH stands at 5.6, whereas it is 6.0 for R-CoOOH, indicating the presence of coordinatively unsaturated Co atoms in S-CoOOH, which is similar to our previously reported nanoribbon Ni(OH)₂. (*Energy Environ. Sci.* 2020, 13, 229-237) Next, based on the optimized model of S-CoOOH structure (Fig.R2-3a), the atomic magnetic moments of coordinatively saturated and unsaturated Co atoms are studied (Fig.R2-3b). For R-CoOOH model, non-magnetic properties could be observed for all Co atoms. In contrast, for S-CoOOH model, the coordinatively unsaturated Co atoms at the edge exhibit ferromagnetism property, with all calculated atomic magnetic moments aligned in the same direction, characteristic of ferromagnetism. Meanwhile, the fully coordinatively saturated Co atoms in the bulk of S-CoOOH remain non-magnetic.

The emergence of ferromagnetic property in S-CoOOH is due to unpaired electrons in high-spin state Co³⁺ ions. In the low-spin state, Co³⁺ demonstrates a fully paired electron configuration, leading to a complete cancellation of magnetic moments and resulting in a non-magnetic behavior at the ground state. In contrast, the high-spin state Co³⁺ configuration shows unpaired electrons, contributing to a net magnetic moment and exhibiting magnetic properties.

Meanwhile, according to the reviewer's advice, we also conducted scanning transmission electron microscopy (STEM) measurements. Fig.R2-4b shows a density of needle-like nano structures. This is similar to the previously reported nanoribbon (NR) structure Ni(OH)₂ (Fig.R2-4a). (*Energy Environ. Sci.* 2020, 13, 229-237) It further confirms the rationality of our optimized S-CoOOH model. However, it should be noted that CoOOH is highly electron beam sensitive, and under an electron beam it would transform into Co₃O₄ as shown in Fig.R2-5. The interlayer spacings are about 0.443 nm and 0.182 nm, well matching the (111)

and $(\bar{1}\bar{1}4)$ planes of Co_3O_4 .

Table R2-2. FT-EXAFS fitting results of R-CoOOH and S-CoOOH, where CN is the coordination number, σ^2 is the Debye-Waller factor.

Sample	path	CN	σ^2	ΔE_0	R	R-factor
R-CoOOH	Co-O	6	$0.0028 \pm$	1.19 (1.29)	$1.903 \pm$	0.006
			0.0012		0.010	
	Co-Co	6	$0.0044 \pm$	-0.47 (1.27)	$2.848 \pm$	
			0.0009		0.008	
S-CoOOH	Co-O	5.6 ± 0.6	$0.0026 \pm$	1.96 (1.98)	$1.910 \pm$	0.007
			0.0012		0.010	
	Co-Co	4.9 ± 0.6	$0.0049 \pm$	-2.05 (2.1)	$2.839 \pm$	
			0.0010		0.009	
Co(OH) ₂ (II)	Co-O	6.1 ± 0.6	$0.0060 \pm$	1.69 (1.26)	$2.093 \pm$	0.007
			0.0009		0.007	
	Co-Co	6.3 ± 0.7	$0.0080 \pm$	3.52 (1.10)	$3.188 \pm$	
			0.0008		0.007	
LiCoO ₂ (III)	Co-O	4.8 ± 0.3	$0.0024 \pm$	3.01 (0.90)	$1.918 \pm$	0.001
			0.0005		0.004	
	Co-Co	5.7 ± 0.2	$0.0028 \pm$	1.21 (0.46)	$2.817 \pm$	
			0.0003		0.002	

Fig.R2-3 a Schematic of optimized model of S-CoOOH structure. b The distribution of magnetization

obtained from DFT, where the inset is *d*-electron configuration of cobalt cations in different spin states at the edge and in the bulk.

Fig.R2-4 **a** STEM HAADF image of NR-Ni(OH)₂ (*Energy Environ. Sci.* 2020, 13, 229-237). **b** Atomically-resolved STEM-HAADF (Scanning Transmission Electron Microscopy - High-Angle Annular Dark Field) image of S-CoOOH.

Fig.R2-5 Atomically-resolved STEM-ABF (Scanning Transmission Electron Microscopy - Annualr Bright Field) image of S-CoOOH.

To address the comment, we have incorporated the proposed modifications into our manuscript (line 194-197, page 10, highlighted in yellow): Additionally, the Scanning Transmission Electron Microscopy (STEM) image of S-CoOOH (Supplementary Fig. S13) shows a density of needle-like nano structures. The above experimental result is similar to our previously reported nanoribbon structure Ni(OH)₂²².

Supplementary Fig. S13 **a** Atomically-resolved STEM-HAADF (Scanning Transmission Electron Microscopy - High-Angle Annular Dark Field) image of S-CoOOH. **b** Atomically-resolved STEM-ABF (Scanning Transmission Electron Microscopy - Annular Bright Field) image of S-CoOOH.

In addition, we have made further revisions in another section of our manuscript (line 213-219, page 10, highlighted in yellow): Next, the atomic magnetic moments of saturated and unsaturated Co atoms are studied based on DFT simulations (Fig. 4b). For R-CoOOH model, non-magnetic property could be observed for all Co atoms. In contrast, for S-CoOOH model, it shows that the Co atoms at edge sites with four-coordination would exhibit ferromagnetism property, with all calculated atomic magnetic moments aligned in the same direction. Conversely, the Co atoms in the bulk of S-CoOOH with six-coordination display non-magnetism. Therefore, the magnetism should be ascribed to the unsaturated coordination at the edge sites in S-CoOOH.

4. The S-CoOOH show very good electrochemical stability. Is there any morphological or structural changes after 200 hours operation?

Response: We are very grateful for the referee's valuable suggestions. According to the reviewer's advice, we have conducted SEM and XAS characterizations for S-CoOOH after 200 h electrochemical oxidation treatment to study the morphological and structural changes. As shown in Fig.R2-6, the morphology of S-CoOOH does not show apparent change after 200 h electrochemical operation. Meanwhile, both Co *K*-edge XAS and FT-EXAFS spectra are kept nearly the same before and after the stability test, further confirming the highly stable structure of S-CoOOH (Fig.R2-7).

Fig.R2-6 SEM images of S-CoOOH, taken before (a) and after (b) a 200-hour stability test. (Supplementary Fig. S26 in the Supplementary Information)

Fig.R2-7 a Normalized Co *K*-edge XAS spectra of S-CoOOH before and after chronopotentiometry (CP) operation for 200 h. b Normalized Co *K*-edge Fourier transformed extended X-ray near fine structure (FT-EXAFS) spectra of S-CoOOH before and after CP operation for 200 h.

To address the comment, we have incorporated the proposed modifications into our manuscript (line 282-287, page 13, highlighted in yellow): Chronopotentiometry (CP) measurement reveals that S-CoOOH exhibits excellent catalytic stability, with negligible decay over approximately 200 h at 10 mA cm^{-2} (Fig. 5g and Supplementary Fig. S25). Furthermore, there is no noticeable change in the morphology of S-CoOOH observed before and after the stability test (Supplementary Fig. S26). Additionally, both Co *K*-edge XAS and FT-EXAFS spectra remain nearly identical for pristine S-CoOOH and that subjected to the 200 h CP test (Fig. 5h, i). These results provide strong evidence for the high structural stability of S-CoOOH.

Supplementary Fig. S26 SEM images of S-CoOOH, taken before (a) and after (b) a 200-hour stability test.

5. I am wondering whether electron doping in R-CoOOH using tetravalent or pentavalent cations (e.g. Ta) can also stabilize high spin state of Co, because the extra electron will occupy e_g state.

Response: We are very grateful for the referee's comment. We agree that extra electrons would occupy e_g state after doping tetravalent or pentavalent cations (e.g., Ta) into the CoOOH lattice, which may be beneficial for stabilizing the high-spin state of Co^{3+} . However, until now it has been quite challenging to stabilize the tetravalent or pentavalent dopants (e.g. Ta) in CoOOH lattice, since such material would often experience a reconstruction process under anodic potential and finally leach out into the electrolyte (*Adv. Energy Mater.* 2023, 13, 2301391; *Nano Lett.* 2023, 23, 5027-5034). In our future work, we would explore a new synthesis method that could successfully incorporate the tetravalent or pentavalent dopants (e.g. Ta) into the CoOOH to study the effect on stabilizing the high-spin state of Co^{3+} .

Reviewer #3

Remarks to the Author: Zhang et al reported that the electrochemical oxidation-derived CoOOH nanoparticles, labeled as S-CoOOH in this manuscript, exhibited a significant enhancement of OER activity, compared to the bulk CoOOH (labeled as R-CoOOH). Based on the results from magnetic analysis, EPR, XAFS, and DFT calculations, the authors attributed this enhancement to the spin state transition of Co ions in S-CoOOH, from low-spin (LS) to high-spin (HS), which was induced by the decrease in the coordination number of Co ions. Although the topic raised in this work, i.e, the role of spin state on the catalysis, is interesting, several issues should be addressed before publication as following:

1. The authors claimed that the CoOOH sample with HS-state Co ions was synthesized for the first time. However, similar result can be found in previous works (for example, *Angew. Chem.* 2015, 127, 8846–8851). Moreover, the existence of HS-state Co ions has been frequently reported in nanostructured cobalt oxides (see *Nat. Commun.* 2016, 7, 11510; *Angew. Chem. Int. Ed.* 2023, e202216837).

Response: We are very grateful for the referee's comment. After carefully reading the literatures provided by the referee, we find that they are not the CoOOH samples with HS-state Co ions. In the following section, these works are discussed in detail.

In this work (*Angew. Chem. Int. Ed.* 2015, 127, 8846-8851), the authors presented an atomically thin CoOOH nanosheet as an efficient electrocatalyst for water oxidation. It was proposed that the excellent OER activity could be ascribed to the change of the valence electron configuration of Co 3d states, leading to effective H₂O molecules absorption and lower OER barrier. The 3d state configuration of the CoOOH nanosheet is shown in Fig.R3-1. Here, it should be noted that the proposed 3d state in this work is not high-spin state Co³⁺ configuration. (The 3d orbital configuration of high-spin state Co³⁺ is provided in Fig.R3-2.)

Fig.R3-1 Electronic structure transformation of ultrathin γ -CoOOH nanosheet and model of water oxidation on the nanosheet surface. (*Angew. Chem. Int. Ed.* 2015, 127, 8846-8851)

Fig.R3-2 The 3d orbital configuration of high-spin state Co^{3+} .

In these two works (*Nat. Commun.* 2016, 7, 11510; *Angew. Chem. Int. Ed.* 2023, 62, e202216837), we agree that high-spin state Co ions could be realized in nanostructured cobalt oxides. The first paper reported a size-induced spin-state transition in LaCoO_3 , where Co^{3+} ions shift from LS to HS, closely approaching the optimal configuration of $e_g^{1,2}$ for OER. (*Nat. Commun.* 2016, 7, 11510) The second paper offered a thorough review of the impact of spin state on catalytic activity (OER and ORR), characterization techniques of spin-state, and methods for spin state modulation, but all in transition metal oxides. (*Angew. Chem. Int. Ed.* 2023, 62, e202216837)

However, most of these transition metal-based oxides are not the real catalytic species, and they would transform into transition metal oxyhydroxides, which act as the real catalytic species for oxygen evolution. (refer to *Nat. Catal.* 2019, 2, 763-772; *Chem. Soc. Rev.* 2021, 50, 8428-8469). Whether high-spin

configuration could be preserved in CoOOH, the actual catalytic species, has not been explored in these papers and needs further consideration.

Different from the previous works, our work for the first time provides the CoOOH with high-spin state Co^{3+} , which is verified by combinative approaches such as electron paramagnetic resonance (EPR), superconducting quantum interference device (SQUID), and X-ray absorption spectroscopy (XAS). The combination of simulation and coordination field theory reveals that the appearance of the high-spin state Co^{3+} is attributed to the coordinatively unsaturated Co atoms at the edge of CoOOH. Moreover, it is found that electron transfer occurs in apex-to-apex e_g^* orbital for high-spin state CoOOH, in contrast to face-to-face t_{2g}^* orbital for low-spin state CoOOH. This high-spin state Co^{3+} electron configuration promotes the electron transfer from electrocatalyst to external circuit via accelerating the deprotonation process, and hence improves intrinsic OER performance.

According to the referee's advice, these mentioned references have been added to our revised manuscript (please see references 7, 16, 17, highlighted in yellow). Also, we have added the following sentences in our revised manuscript (line 47-50, page 2, highlighted in yellow): Although the introduction of high-spin state Co ions has been reported in Co-based oxides¹⁶⁻¹⁹, most of them would experience an irreversible reconstruction process under anodic alkaline conditions, forming cobalt oxyhydroxides (CoOOH), which act as the actual catalytic species for oxygen evolution^{20,21}.

2. For nanostructured metal oxides, the decrease in the coordination for metal ions is usually associated with the presence of oxygen vacancies. Consequently, the valence state of metal ions would be reduced. As shown in Fig. 3b, a red shift in the Co K-edge X-ray absorption suggested a lowered Co valence state in S-CoOOH. It should notice that both the Co valence state and spin state can regulate the e_g orbital configuration. Therefore, to explore the role of spin state, the author should exclude the contribution from the reduction in the Co valence state.

Response: We are very grateful for the referee's comment. We agree that both Co valence state and spin state can regulate the e_g orbital configuration. Generally, the valence state of Co is determined by a combination

of several factors, *i.e.* hydrogen coverage and Co-O coordination number. And, the regulation of e_g orbital configuration by Co valence states is usually ascribed to changes in hydrogen coverage.

In our work, the hydrogen coverage of S-CoOOH model, remains consistent with R-CoOOH, preserving a stable Co:O:H ratio of 1:1:2. Hence, the slight variation of Co valence is mainly induced by the Co-O coordination. As shown in Fourier transformed extended X-ray near fine structure (FT-EXAFS) fitting results, the fitted Co-O coordination number changes from 6.0 in R-CoOOH to 5.6 in S-CoOOH (Table R3-1). As a result, in our work the variation of Co valence has negligible influence on the e_g orbital configuration.

Table R3-1. FT-EXAFS fitting results of R-CoOOH and S-CoOOH, where CN is the coordination number, σ^2 is the Debye-Waller factor.

Sample	path	CN	σ^2	ΔE_0	R	R-factor
R-CoOOH	Co-O	6	0.0028 ±	1.19 (1.29)	1.903 ±	0.006
			0.0012		0.010	
	Co-Co	6	0.0044 ±	-0.47 (1.27)	2.848 ±	
			0.0009		0.008	
S-CoOOH	Co-O	5.6 ± 0.6	0.0026 ±	1.96 (1.98)	1.910 ±	0.007
			0.0012		0.010	
	Co-Co	4.9 ± 0.6	0.0049 ±	-2.05 (2.1)	2.839 ±	
			0.0010		0.009	
Co(OH) ₂ (II)	Co-O	6.1 ± 0.6	0.0060 ±	1.69 (1.26)	2.093 ±	0.007
			0.0009		0.007	
	Co-Co	6.3 ± 0.7	0.0080 ±	3.52 (1.10)	3.188 ±	
			0.0008		0.007	
LiCoO ₂ (III)	Co-O	4.8 ± 0.3	0.0024 ±	3.01 (0.90)	1.918 ±	0.001
			0.0005		0.004	
	Co-Co	5.7 ± 0.2	0.0028 ±	1.21 (0.46)	2.817 ±	
			0.0003		0.002	

We have added the following sentences in our revised manuscript (line 206-212, page 10, highlighted in yellow): It should be noticed that both Co valence state change induced by hydrogen coverage and spin state could regulate the e_g^* orbitals configuration. Based on S-CoOOH and R-CoOOH models, we find that the hydrogen coverage of S-CoOOH remains consistent with R-CoOOH, preserving a Co:O:H ratio of 1:1:2 (seen in the CONTCAR of optimized models). This excludes the possible contribution from the variation of Co valence induced by hydrogen coverage on the e_g^* orbital configuration. Hence, the slight decrease of Co valence state is associated with the appearance of high-spin state Co^{3+} .

3. From the XAFS fitting results (Fig. S12 and Table S2), the Co-O coordination number in S-CoOOH was 5.7, just a little lower than that in R-CoOOH (6). This means that the proportion of coordinatively unsaturated Co atoms in S-CoOOH is small. Moreover, the S-CoOOH structure model as shown in Fig. 4a illustrated that the coordinatively unsaturated Co atoms only appeared at edge sites. Those results suggest that even if the coordinatively unsaturated Co atoms were in HS state as the authors stated, the proportion of them would be small in S-CoOOH. The author should clearly illustrate this point in the manuscript.

Response: We deeply appreciate the referee's insightful suggestion. We agree with the reviewer's comment that the number of coordinatively unsaturated Co atoms in HS state is very small, as compared with the coordinatively saturated Co atoms. This small number of coordinatively unsaturated Co atoms in HS state makes the catalytic property of CoOOH much higher than that of R-CoOOH. Motivated by this, more efforts could be directed towards increasing the quantity of high-spin state Co^{3+} sites, in order to further enhance the OER activity.

To address the comment, we have incorporated the proposed modifications into our manuscript (line 277-281, page 13, highlighted in yellow): It is noticed that the number of coordinatively unsaturated Co atoms in the high-spin state is very small, as compared with the coordinatively saturated Co atoms (Fig. 4a). Yet, the small number of coordinatively unsaturated Co atoms in the high-spin state makes the catalytic property of CoOOH much higher than that of R-CoOOH, demonstrating the superior activity of the high-spin state Co^{3+} atoms.

4. For R-CoOOH, the authors stated that it would be paramagnetic at high temperature (300 K) and antiferromagnetic at low temperature (3 K). Ideally, both the paramagnetic and antiferromagnetic states give rise to a linear dependence of magnetization on the magnetic field. However, in Fig. S9, the magnetic hysteresis loops at 3 K for R-CoOOH exhibited a non-linear behavior. Why? On the other hand, for S-CoOOH, the magnetic hysteresis loops at 3 K revealed that both the remanent magnetization and the coercive field were reduced to zero. Why? Please explain the above results.

Response: We are very grateful for the referee's comment. The nonlinear behavior of the magnetic hysteresis (M-H) loop at 3 K for R-CoOOH might be due to the following reasons:

1. The decreased antiferromagnetic coupling effect. The M-H loop measurement was conducted at 3 K, which was close to the Néel temperature (around 10 K, Fig.R3-3). Near the Néel temperature, the antiferromagnetic coupling would weaken, leading to a non-linear shift. Such a phenomenon of anomalous non-linear M-H loops near the Néel temperature has been reported in many antiferromagnetic materials, *i.e.* EuNiGe₃, C₁₃H₁₃Ba₄ClCo₃O₂₆ (Fig.R3-4). (*J. Phys. Chem. Kett.* 2023, 14, 1000-1006; *Inorg. Chem.* 2022, 61, 2265-2271)

2. A spin-flop transition. The spin-flop transition generally occurs in antiferromagnetic materials that have weak magnetic anisotropy. As shown in Fig.R3-5, Lee et al. proposed that the non-linear deviation observed in the antiferromagnetic material Co₄Ta₂O₉ was attributed to a spin-flop transition at $H_c \approx 0.3$ T for an applied field. (*Sci. Rep.* 2020, 10, 12362).

Fig.R3-3 ZFC (zero field cooled) and FC (field cooled) magnetization for R-CoOOH and S-CoOOH as a function of temperature with applied magnetic field $H = 100$ Oe.

Fig.R3-4 a Temperature dependence of the magnetic moment of Eu at ambient pressure. The line presents a plot of the Brillouin function for parameters characteristic to the divalent Eu. Inset: AFM-type hysteresis loop measured at 8 K. **b** M-H loops and its derivative recorded at 2K for $C_{13}H_{13}Ba_4ClCo_3O_{26}$. (Ref. to *J. Phys. Chem. Lett.* 2023, 14, 1000-1006; *Inorg. Chem.* 2022, 61, 2265-2271)

Fig.R3-5 a Isothermal magnetization M measured at 2 K along the a , b^* , and c axes. **b** Magnified plot of M for the a direction. **c** Antiferromagnetic spin structure at $H = 0$ T (top), and magnetic structure above the spin-flop transition, $H > H_c$ along the a axis (bottom). (Ref. to *Sci. Rep.* 2020, 10, 12362)

For S-CoOOH, why both the remanent magnetization and the coercive field were reduced to zero might be due to the following possible reasons. As discussed in Question 3, there is only a quite small number of high-

spin state Co^{3+} atoms, which contributes fixed magnetic moments and ferromagnetic signals in S-CoOOH. At low temperatures, the paramagnetic contribution from the low-spin state Co^{3+} might dominate the whole magnetic phenomenon, which would finally result in the negligible remanent magnetization and coercive field observed at 3 K. This would be further explored in our further work.

To address the comment, we have incorporated the proposed modifications into our manuscript (line 126-133, page 6, highlighted in yellow): Further analysis of the temperature-dependent magnetization (M-T) curves for R-CoOOH (Fig. 2c) reveals an inflection point around 10 K, referred as the Néel temperature, suggesting a transition from a paramagnetic to an antiferromagnetic state. Considering that 3 K is below the Néel temperature, it confirms antiferromagnetic behavior of R-CoOOH at 3K (Supplementary Fig. S6), where all electrons are paired up and adjacent valence electrons have opposite spin directions. The non-linear antiferromagnetic character in the M-H loop of R-CoOOH might be due to weak antiferromagnetic coupling near the Néel temperature^{29,30} or a spin-flop transition influenced by weak magnetic anisotropy³¹.

In addition, we have also added a detailed discussion below Supplementary Fig. S6 in the Supplementary Information as follows: Supplementary Fig. S6 is the magnetic hysteresis (M-H) loops of R-CoOOH and S-CoOOH recorded at 3 K. The nonlinear M-H loop at 3 K for R-CoOOH might be due to the following reasons: 1. The decreased antiferromagnetic coupling effect. The M-H loop measurement was conducted at 3 K, which was close to the Néel temperature (around 10 K, Fig. 2d). Near the Néel temperature, the antiferromagnetic coupling would weaken, leading to a non-linear shift. Such a phenomenon of anomalous non-linear M-H loops near the Néel temperature has been reported in many antiferromagnetic materials, *i.e.* EuNiGe_3 ⁶, $\text{C}_{13}\text{H}_{13}\text{Ba}_4\text{ClCo}_3\text{O}_{26}$ ⁷. 2. A spin-flop transition. The spin-flop transition generally occurs in antiferromagnetic materials that have weak magnetic anisotropy. Lee et al. proposed that the non-linear deviation observed in the antiferromagnetic material $\text{Co}_4\text{Ta}_2\text{O}_9$ was attributed to a spin-flop transition at $H_c \approx 0.3$ T for an applied field⁸. For S-CoOOH, why both the remanent magnetization and the coercive field are reduced to zero might be due to the following possible reasons. There is only a quite small number of high-spin state Co^{3+} atoms, which contributes fixed magnetic moments and ferromagnetic signals in S-CoOOH. At low temperatures, the paramagnetic contribution from the low-spin state Co^{3+} might dominate the whole magnetic phenomenon, which would finally result in the negligible remanent magnetization and

coercive field observed at 3 K. This would be further explored in our further work.

5. In Fig. 2c, the ZFC and FC magnetization curves diverge for both R-CoOOH and S-CoOOH. Why?

Response: We are very grateful for the referee's comment. For R-CoOOH, it exhibits paramagnetic behavior above the Néel temperature and antiferromagnetic behavior below the Néel temperature. Due to the applied magnetic field of 100 Oe for the FC curve, the relatively weak moment was induced, hence the slight divergence between the ZFC and FC curves was observed above the Néel temperature (Fig.R3-6). In addition, a significant divergence between ZFC and FC magnetization curves near the Néel temperature (about 10 K) is attributed to the transition between paramagnetic and antiferromagnetic properties.

For S-CoOOH, the slight divergence between ZFC and FC curves in the high-temperature region (~100-300 K) may be due to the small coercive field of S-CoOOH. This hypothesis could be verified by the M-H loop of S-CoOOH at 300 K (Fig.R3-7a). The moment at 0.01 T (100 Oe, indicative of FC at 300 K) is slightly higher than at 0 T (indicative of ZFC at 300 K) (Fig.R3-7b). An applied magnetic field of 100 Oe, which is near the coercive field and will tilt moment along magnetic field direction, might lead to a slight divergence between ZFC and FC curves. Moreover, the increased trend noted in the FC at low temperatures (~3-50 K) can be attributed to the paramagnetic contribution from the low-spin state Co^{3+} , similar to behaviors observed in the previously reported partially high-spin state LaCoO_3 (Fig.R3-8). (*Nat. Commun.* 2016, 7, 11510)

Fig.R3-6 ZFC (zero field cooled) and FC (field cooled) magnetization for R-CoOOH and S-CoOOH as a function of temperature with applied magnetic field $H = 100$ Oe.

Fig.R3-7 a Magnetic hysteresis loop of S-CoOOH, recorded at 300 K. b The enlarged view of Figure a, between -0.10 and 0.10 T (-1000 Oe and 1000 Oe). (Supplementary Fig. S7 in the Supplementary Information)

Fig.R3-8 Magnetic properties of the bulk and nanosized LCO. Temperature dependent magnetization under $H = 1$ kOe. The spin states are estimated to be 64.7% HS + 36.3% LS, 60.5% HS + 39.5% LS, 55% HS + 45% LS, and 50% HS + 50% LS for the 60, 80, 200 nm and bulk samples. (*Nat. Commun.* 2016, 7, 11510)

To address the comment, we have incorporated the proposed modifications into our manuscript (line 134-136, page 6, highlighted in yellow): Moreover, the slight divergences between ZFC (zero field cooled) and FC (field cooled) curves of both R-CoOOH and S-CoOOH are discussed in Supplementary Fig. S7.

A detailed discussion has been added to the Supplementary Information following Supplementary Fig. S7: The slight divergences between ZFC (zero field cooled) and FC (field cooled) curves of R-CoOOH and S-CoOOH could be explained as below: For R-CoOOH, it exhibits paramagnetic behavior above the Néel temperature and antiferromagnetic behavior below the Néel temperature. Due to the applied magnetic field of 100 Oe for the FC curve, the relatively weak moment is induced, hence the slight divergence between the ZFC and FC curves is observed above the Néel temperature (Fig. 2c). In addition, a significant divergence between the ZFC and FC curves near the Néel temperature (about 10 K) is attributed to the transition between paramagnetic and antiferromagnetic properties. For S-CoOOH, the slight divergence between the ZFC and FC curves in the high-temperature region (~ 100 -300 K) may be due to the small coercive field of S-CoOOH. This hypothesis could be verified by the M-H loop of S-CoOOH at 300 K (Supplementary Fig. S7a). The moment at 0.01 T (100 Oe, indicative of FC at 300 K) is slightly higher than at 0 T (indicative of ZFC at 300 K) (Supplementary Fig. S7b). An applied magnetic field of 100 Oe, which is near the coercive field and will

tilt moment along magnetic field direction, might lead to a slight divergence between the ZFC and FC curves. Moreover, the increased trend noted in the FC at low temperatures (~3-50 K) can be attributed to the paramagnetic contribution from the low-spin state Co^{3+} , similar to behaviors observed in the previously reported partially high-spin state LaCoO_3 ⁹.

Supplementary Fig. S7 **a** Magnetic hysteresis (M-H) loop of S-CoOOH, recorded at 300 K. **b** The enlarged result of Figure a, between -0.10 and 0.10 T (-1000 Oe and 1000 Oe).

6. The magnetic moments and interactions would give evidence of spin configuration, which can be obtained by fitting the temperature-dependent magnetizations via the Curie-Weiss law. This fitting is suggested to be carried out for both samples.

Response: We are very grateful for the referee's comment. According to the reviewer's advice, we used the method provided in the papers recommended by the referee above, to conduct the fitting of the magnetic moments and interactions. (*Nat. Commun.* 2016, 7, 11510; *Angew. Chem. Int. Ed.* 2023, 62, e202216837) The temperature-dependent magnetizations were measured with a magnetic field of $H = 100$ Oe under field-cooling procedures for R-CoOOH (Fig.R3-9a). In the paramagnetic region, above 10 K for R-CoOOH, the susceptibilities derived from the magnetizations ($\chi = M/H$) obey a paramagnetic Curie-Weiss law: ($\chi = C/(T - T_c)$ where C is Curie constant, and T_c is Curie-Weiss temperature. From the fitting result (Fig.R3-9b), the effective magnetic moment μ_{eff} could be calculated through the equation $\mu_{eff} = \sqrt{8C}\mu_B$, where μ_B is Bohr magneton. Here, the calculated μ_{eff} for R-CoOOH is $0.09 \mu_B$, which is close to the theoretical predictions from DFT simulations and reported values for low-spin state Co^{3+} in the literatures. (*Nat. Commun.* 2016, 7, 11510; *Angew. Chem. Int. Ed.* 2023, 62, e202216837)

Fig.R3-9 Magnetic property of R-CoOOH. a Temperature dependent magnetization under $H = 100$ Oe. **b** The temperature dependence inverse susceptibility. The dotted line is the fitting result by a Curie-Weiss law. (Supplementary Fig. S8 in the Supplementary Information)

However, S-CoOOH exhibits ferromagnetic behavior at 300 K, with a Curie temperature above 300 K. The Curie-Weiss law is not well applicable within the ferromagnetic region. (Refer to *Nat. Commun.* 2016, 7, 11510; *Angew. Chem. Int. Ed.* 2023, 62, e202216837) Also, given the mixed ferromagnetic and paramagnetic properties for S-CoOOH, the utilization of the Curie-Weiss law in determining the magnetic moments and interactions would be challenging as the temperature dependency of ferromagnetic and paramagnetic behaviors are significantly different. On the other hand, the ferromagnetic property of S-CoOOH is contributed by the edge coordinatively unsaturated Co atoms, which only constitute a quite small fraction of the overall structure.

Moreover, we investigate the spin configuration of S-CoOOH and R-CoOOH via a combination of various characterizations *i.e.* magnetic hysteresis (M-H) loops, electron paramagnetic resonance (EPR) spectra, Co *K*-edge, Co *L*-edge, O *K*-edge X-ray absorption spectroscopy (XAS), and density functional theory (DFT) simulations, as detailed below:

- 1. M-H loops recorded at room-temperature (300 K)**, which is the actual experimental condition for OER, the dominant ferromagnetic behavior could be inferred by the M-H loops of S-CoOOH (Fig.R3-10a). In contrast, the M-H loop of R-CoOOH reveals paramagnetic characteristic (Fig.R3-10b,c).
- 2. EPR spectra (Fig.R3-10d)**, a distinct peak with a *g* value of 2.15 is observed for S-CoOOH, but not for

R-CoOOH. This peak is attributed to high-spin state Co^{3+} , showcasing ferromagnetism characteristic.

3. Co *K*-edge XAS spectra (Fig.R3-11a), the noticeable reduction in the white line intensity (Fig.R3-11b) demonstrates that the $4p$ orbitals in S-CoOOH sample experience splitting, indicating the existence of high-spin state Co^{3+} . Moreover, the diminished intensity of the pre-edge and the elongation of Co-O bond length also serve as evidence of the presence of high-spin state Co^{3+} in S-CoOOH (Fig.R3-11c and d).

4. Co *L*-edge XAS spectra (Fig.R3-12a), both L_3 and L_2 peaks of S-CoOOH become broader with lower intensity than those of R-CoOOH. This phenomenon also suggests that the $3d$ band becomes broader, which could be ascribed to the high-spin state Co^{3+} .

5. O *K*-edge XAS spectra (Fig.R3-12b), S-CoOOH shows a lower intensity around 531 to 533 eV (O $1s$ to O $2p$ -Co $3d$ hybrid orbitals), accompanied by more pronounced peak splitting (the inset of Fig.R3-12b), indicating more splitting of $3d$ orbitals. It suggests that S-CoOOH is in a high-spin state.

6. DFT simulations were performed to calculate the atomic magnetic moment of S-CoOOH, with the optimized structure model (Fig.R3-13). It is found that the coordinatively unsaturated Co atoms at the edge exhibit ferromagnetism properties. This is evidenced by the fact that all calculated atomic magnetic moments are positive, indicating that the magnetic moments are aligned in the same direction, characteristic of ferromagnetism.

Fig.R3-10 Magnetic analysis of R-CoOOH and S-CoOOH samples. **a-b** Magnetic hysteresis loop of R-CoOOH (**a**) and S-CoOOH (**b**) at 300 K, where the insets are magnetic ordering patterns of Co^{3+} ions in low-spin and high-spin states. **c** ZFC (zero field cooled) and FC (field cooled) magnetization curves for R-CoOOH and S-CoOOH as a function of temperature with applied magnetic field $H = 100$ Oe. **d** EPR spectra of R-CoOOH and S-CoOOH recorded at 300 K.

Fig.R3-11 **a** Normalized Co K-edge XAS spectra of both R-CoOOH and S-CoOOH. **b** The enlarged result within white line around 7727 to 7732 eV, extracted from **a**. **c** The enlarged pre-edge regions of Co K-edge spectra, extracted from **b**. **d** Normalized Co K-edge Fourier transformed extended X-ray near fine structure (FT-EXAFS) comparison between R-CoOOH and S-CoOOH, extracted from **b**.

Fig.R3-12 **a** Normalized Co *L*-edge XAS spectra of R-CoOOH and S-CoOOH. **b** Normalized O *K*-edge spectra of R-CoOOH and S-CoOOH (the inset shows the enlarged result of S-CoOOH within pre-edge around 531 to 533 eV).

Fig.R3-13 Magnetism distribution of R-CoOOH and S-CoOOH. **a** Schematic of optimized model of S-CoOOH. **b** The distribution of magnetization obtained from DFT, where the inset is *d*-electron configuration of cobalt cations in different spin states at the edge and in the bulk.

To address the comment, we have incorporated the proposed modifications into our manuscript (line 136-139, page 6, highlighted in yellow): Additionally, the M-T measurements (Fig.2c) reveal that, in the paramagnetic region above 10 K for R-CoOOH, the calculated effective magnetic moment μ_{eff} is $0.09 \mu_B$ (Supplementary Fig. S8), which is close to those values reported for low-spin state Co^{3+} ^{16,17}.

A detailed discussion has been added to the Supplementary Information following Supplementary Fig. S8: The temperature-dependent magnetizations (M-T) were measured with a magnetic field of $H=100$ Oe under field-cooling procedures for R-CoOOH (Supplementary Fig. S8a). In the paramagnetic region, above 10 K for R-CoOOH, the susceptibilities derived from the magnetizations ($\chi = M/H$) obey a paramagnetic Curie-Weiss law: ($\chi = C/(T - T_c)$) where C is Curie constant, and T_c is Curie-Weiss temperature. From the fitting results (Supplementary Fig. S8b), the effective magnetic moment μ_{eff} could be calculated through the equation $\mu_{eff} = \sqrt{8C}\mu_B$, where μ_B is Bohr magneton. Here, the calculated μ_{eff} for R-CoOOH is $0.09 \mu_B$, which is close to the theoretical predictions from DFT simulations and reported values for low-spin state Co^{3+} in the literatures^{9,10}. However, S-CoOOH exhibits ferromagnetic behavior at 300 K, with a Curie temperature above 300 K. The Curie-Weiss law is not applicable within the ferromagnetic region. Also, given the mixed ferromagnetic and paramagnetic properties for S-CoOOH, the utilization of the Curie-Weiss law in determining the magnetic moments and interactions would be challenging as the temperature dependency of ferromagnetic and paramagnetic behaviors are significantly different. On the other hand, the ferromagnetic property of S-CoOOH is contributed by the edge coordinatively unsaturated Co atoms, which only constitute a quite small fraction of the overall structure.

Supplementary Fig. S8 Magnetic property of R-CoOOH. **a** Temperature dependent magnetization (M-T) under $H = 100$ Oe. **b** The temperature dependence inverse susceptibility. The dotted line is the fitting result by a Curie-Weiss law.

7. In Co K-edge XAS spectra (Fig. 3b), the pre-edge peak around 7710 eV corresponds to the Co 1s-3d

transition, which would evidence the spin configuration of Co ions. A detailed analysis on this peak is suggested. Additionally, Co L-edge XAS is also suggested to be carried out to get more evidence of the spin state transition.

Response: We are very grateful for the referee's insightful suggestions. In Co *K*-edge XAS spectra, the pre-edge peak around 7710 eV corresponds to the Co *1s-3d* transition, and its intensity is correlated to the centrosymmetry of the octahedron in CoOOH. A higher pre-edge peak represents a lower degree of centrosymmetry. For the high-spin state Co³⁺, the *3d* band is broader than the low-spin state Co³⁺, indicating a lower degree of centrosymmetry. Fig.R3-14a shows that the pre-edge peak intensity of S-CoOOH is higher than that of R-CoOOH, which is consistent with our proposal.

Meanwhile, according to the reviewer's advice, we have further carried out Co *L*-edge XAS measurements to study the spin state transition. As depicted in Fig.R3-14b the peak corresponding to high spin at around 780 eV of S-CoOOH is higher than that of R-CoOOH, indicating the emergence of high-spin state Co³⁺. (*Chem. Mater.* 2022, 34, 10509-10516) Moreover, both *L*₃ and *L*₂ peaks of S-CoOOH become broader with lower intensity than those of R-CoOOH, which suggests that the *3d* band becomes broader. This could also be ascribed to the high-spin state Co³⁺. (*Energy Environ. Sci.* 2023, 16, 641-652.)

Fig.R3-14 a Enlarged Co *K*-edge pre-edge region of R-CoOOH and S-CoOOH, extracted from Fig. 3b. **b** Normalized Co *L*-edge XAS spectra of R-CoOOH and S-CoOOH. (Supplementary Fig. S10 and Supplementary S12 in the Supplementary Information)

To address the comment, we have added these sentences in our manuscript (line 164-169, page 8, highlighted in yellow): Meanwhile, the spin configuration is also analyzed by the pre-edge peak around 7710 eV, corresponding to the Co $1s$ - $3d$ transition, whose intensity of the pre-edge peak is correlated to the centrosymmetry of the octahedron in CoOOH^{18} . As shown in Supplementary Fig. S10, the pre-edge peak intensity of S-CoOOH is higher than that of R-CoOOH, showing a lower degree of centrosymmetry, which further confirms the presence of high-spin state Co^{3+} .

Supplementary Fig. S10 Enlarged Co K -edge pre-edge region of R-CoOOH and S-CoOOH, extracted from Fig. 3b.

In addition, we have made further revisions in another section of our manuscript (line 177-180, page 8, highlighted in yellow): The Co L -edge XAS spectra (Supplementary Fig. S12) show that both L_3 and L_2 peaks of S-CoOOH become broader with lower intensity than those of R-CoOOH, which suggests that the $3d$ band becomes broader, indicating the emergence of the high-spin state Co^{3+18} .

Supplementary Fig. S12 Normalized Co *L*-edge XAS spectra of R-CoOOH and S-CoOOH.

8. The Co-O bond length for HS state should be larger than that for LS state. Please give an analysis on the Co-O bond length from Co *k*-edge XAFS for both the samples.

Response: We are very grateful for the referee's comment. The FT-EXAFS spectra for both R-CoOOH and S-CoOOH, along with the fitting data, are shown in Fig.R3-15 and Table R3-2. It is revealed that the fitted average Co-O bond length is 1.910 Å for S-CoOOH, which is larger than that of R-CoOOH (1.903Å). This elongated Co-O bond length further supports the existence of high-spin state Co^{3+} in S-CoOOH.

Fig.R3-15 a Fitting results for S-CoOOH (the inset is the corresponding coordination number). **b** Normalized Co *K*-edge Fourier transformed extended X-ray near fine structure (FT-EXAFS) comparison between R-CoOOH and S-CoOOH. (Supplementary Fig. S11 in the Supplementary Information)

Table R3-2. FT-EXAFS fitting results of R-CoOOH and S-CoOOH, where CN is the coordination number, σ^2 is the Debye-Waller factor.

Sample	path	CN	σ^2	ΔE_0	R	R-factor
R-CoOOH	Co-O	6	0.0028 ±	1.19 (1.29)	1.903 ±	0.006
			0.0012		0.010	
	Co-Co	6	0.0044 ±	-0.47 (1.27)	2.848 ±	
			0.0009		0.008	
S-CoOOH	Co-O	5.6 ± 0.6	0.0026 ±	1.96 (1.98)	1.910 ±	0.007
			0.0012		0.010	
	Co-Co	4.9 ± 0.6	0.0049 ±	-2.05 (2.1)	2.839 ±	
			0.0010		0.009	
Co(OH) ₂ (II)	Co-O	6.1 ± 0.6	0.0060 ±	1.69 (1.26)	2.093 ±	0.007
			0.0009		0.007	
	Co-Co	6.3 ± 0.7	0.0080 ±	3.52 (1.10)	3.188 ±	
			0.0008		0.007	
LiCoO ₂ (III)	Co-O	4.8 ± 0.3	0.0024 ±	3.01 (0.90)	1.918 ±	0.001
			0.0005		0.004	
	Co-Co	5.7 ± 0.2	0.0028 ±	1.21 (0.46)	2.817 ±	
			0.0003		0.002	

To address the comment, we have added these sentences in the revised manuscript as detailed below (line 169-173, page 8, highlighted in yellow): The average Co-O bond length is analyzed via fitting the normalized Co *K*-edge Fourier transformed extended X-ray near fine structure (FT-EXAFS) spectra (Supplementary Fig. S11 and Supplementary Table S2). The results show that the fitted Co-O bond length is 1.910 Å for S-CoOOH, longer than that of R-CoOOH (1.903Å). This elongated Co-O bond length further supports the existence of high-spin state Co³⁺ in S-CoOOH.

Supplementary Fig. S11 a Fitting results for S-CoOOH (the inset is the corresponding coordination number). **b** Normalized Co *K*-edge Fourier transformed extended X-ray near fine structure (FT-EXAFS) comparison between R-CoOOH and S-CoOOH.

Supplementary Table S2. FT-EXAFS fitting results of R-CoOOH and S-CoOOH, where CN is the coordination number, σ^2 is the Debye-Waller factor.

Sample	path	CN	σ^2	ΔE_0	R	R-factor
R-CoOOH	Co-O	6	$0.0028 \pm$	1.19 (1.29)	$1.903 \pm$	0.006
			0.0012		0.010	
	Co-Co	6	$0.0044 \pm$	-0.47 (1.27)	$2.848 \pm$	
			0.0009		0.008	
S-CoOOH	Co-O	5.6 ± 0.6	$0.0026 \pm$	1.96 (1.98)	$1.910 \pm$	0.007
			0.0012		0.010	
	Co-Co	4.9 ± 0.6	$0.0049 \pm$	-2.05 (2.1)	$2.839 \pm$	
			0.0010		0.009	
Co(OH) ₂ (II)	Co-O	6.1 ± 0.6	$0.0060 \pm$	1.69 (1.26)	$2.093 \pm$	0.007
			0.0009		0.007	
	Co-Co	6.3 ± 0.7	$0.0080 \pm$	3.52 (1.10)	$3.188 \pm$	
			0.0008		0.007	
LiCoO ₂ (III)	Co-O	4.8 ± 0.3	$0.0024 \pm$	3.01 (0.90)	$1.918 \pm$	0.001
			0.0005		0.004	
	Co-Co	5.7 ± 0.2	$0.0028 \pm$	1.21 (0.46)	$2.817 \pm$	
			0.0003		0.002	

9. Dose the calculated PDOS support that the magnetic ground state of R-CoOOH is antiferromagnetic?

Response: We are very grateful for the referee's comment. In our work, the calculated PDOS does not support the antiferromagnetic ground state of R-CoOOH, which is due to the limitation of calculation. Generally, in view of calculation, the bulk CoOOH displays non-magnetic property, while the surface Co atoms would lean towards an antiferromagnetic alignment, because the energy of an antiferromagnetic surface is lower than that of a non-magnetic one (*Nat. Energy* 2022, 7, 765-773). In our calculation, the PDOS result is also obtained based on the general bulk CoOOH model. As such, it shows a non-magnetic ground state for R-CoOOH.

To address the comment, we have added these sentences in the revised manuscript as detailed below (line 226-229, page 11, highlighted in yellow): It should be noted that, in view of calculation, the bulk CoOOH displays non-magnetic property, while the surface Co atoms in real structure would lean towards an antiferromagnetic alignment, since the energy of an antiferromagnetic surface is lower than that of a non-magnetic one¹⁰.

10. The CV curves (Fig. S17c) for R-CoOOH are not centered at 0 mA. Why?

Response: We are very grateful for the referee's constructive correction. The CV curves might be influenced by the adsorbed bubbles after electrochemical tests. According to the reviewer's advice, we have re-conducted the CV characterization after removing the bubbles and making the necessary updates. The revised figures (Fig.R3-16) are presented below:

Fig.R3-16 The electrochemical surface area (ECSA) of R-CoOOH sample. **a** CV curves of R-CoOOH in 1 M KOH at different scan rates from 10 to 50 mV s^{-1} , charging current densities at 0.65 V versus RHE plotted against the scan rate. **b** Current versus scan rate plot for the estimation of ECSA.

To address the comment, we revised Supplementary Fig. S20 (previous Supplementary Fig. S17) in the Supplementary Information concerning the electrochemical surface area (ECSA) of R-CoOOH, as detailed below:

Supplementary Fig. S20 The electrochemical surface area (ECSA) of S-CoOOH and R-CoOOH samples.

a and c CV curves of S-CoOOH and R-CoOOH in 1 M KOH at different scan rates from 10 to 50 mV s⁻¹, charging current densities at 0.65 V versus RHE plotted against the scan rate. **b and d** Δj versus scan rate plot for the estimation of ECSA.

Moreover, we have also revised the manuscript regarding the electrochemical data associated with the electrochemical surface area (ECSA) of R-CoOOH (line 267-269, page 13, highlighted in yellow): The resulting ECSA for both S-CoOOH and R-CoOOH are 256.25 cm² and 176.50 cm², respectively (Supplementary Fig. S20).

In addition, the figures related to the electrochemical surface area (ECSA) of R-CoOOH have been updated as below, including Fig. 5e and Supplementary Fig. S21 (previous Supplementary Fig. S18):

Fig. 5 OER activity measurements of R-CoOOH and S-CoOOH. **a** PDOS of Co-3d and O-2p orbitals for

R-CoOOH. **b** PDOS of Co-3*d* and O-2*p* orbitals for S-CoOOH. **c** Charge versus potential from pulse-voltammetry (P-V) measurements. **d** OER polarization curves after *iR*-correction. **e** OER polarization curves with current normalized to ECSA after *iR*-correction. **f** OER polarization curves with current normalized to mass loading after *iR*-correction. **g** Chronopotentiometry (CP) operation of S-CoOOH at 10 mA cm⁻² for 200 h. **h** Normalized Co *K*-edge XAS spectra of S-CoOOH before and after CP operation for 200 h. **i** Normalized Co *K*-edge FT-EXAFS spectra of S-CoOOH before and after CP operation for 200 h.

Supplementary Fig. S21 Electrochemical performance of S-CoOOH and R-CoOOH samples without *iR* correction. **a** OER curves of CoOOH based on a backward scan conducted at a scan rate of 0.1 mV s⁻¹. **b** OER polarization curves normalized to ECSA. **c** OER curves normalized to mass loading. **d** Summary of overpotentials at 10 mA cm⁻², current density at 250 mV overpotential, current density normalized to loading mass at 250 mV overpotential.

11. In Fig. S19a, S-CoOOH exhibited two distinct redox peaks but R-CoOOH did not. Why? If the peak

around 1.4 V belongs to the Co III/IV redox process, the OER active sites for S-CoOOH would be Co IV ions. In this case, how does the spin state of Co^{3+} ions influence the OER activity?

Response: We are very grateful for the referee's comment. The distinct redox peaks both appear in the S-CoOOH and R-CoOOH samples, as shown in the enlarged OER curves (Fig.R3-17). According to the previous study (*Nature*, 593, 67–73, 2021), the peak around 1.0 V is ascribed to the $\text{Co}^{2+}/2.5+$ redox process, while the peak around 1.4 V belongs to the $\text{Co}^{2.5+}/3+$ redox process, as shown in Fig.R3-18.

Fig.R3-17 OER curves based on a backward scan conducted at a scan rate of 0.1 mV s^{-1} without iR correction, including detailed enlargements of recorded redox peaks. **a, b** Enlarged view of the OER curves for S-CoOOH. **c, d** Enlarged view of the OER curves for R-CoOOH. (Supplementary Fig. S22 in the Supplementary Information)

Fig.R3-18 Schematic demonstrating the translation and the expansion and contraction of individual CoO_2 layers as the voltage is increased. (Refer. to *Nature*, 2021, 593, 67-74)

To address the comment, we have added these sentences in the revised manuscript as detailed below (line 272-274, page 13, highlighted in yellow): The raw electrochemical data without iR correction, including detailed enlargements of two recorded redox peaks corresponding to $\text{Co}^{2+/2.5+}$ and $\text{Co}^{2.5+/3+}$ redox couples³⁶, are displayed in Supplementary Fig. S21-23.

In addition, we have revised Supplementary Fig. S20 (previous Supplementary Fig. S17) and added Supplementary Fig. S21 in the Supplementary Information as below:

Supplementary Fig. S21 Electrochemical performance of R-CoOOH and S-CoOOH samples without iR correction. **a** OER curves of CoOOH based on a backward scan conducted at a scan rate of 0.1 mV s^{-1} . **b** OER polarization curves normalized to ECSA. **c** OER curves normalized to mass loading. **d** Summary of overpotentials at 10 mA cm^{-2} , current density at 250 mV overpotential, current density normalized to loading mass at 250 mV overpotential.

Supplementary Fig. S22 OER curves based on a backward scan conducted at a scan rate of 0.1 mV s^{-1} without iR correction, including detailed enlargements of recorded redox peaks. **a, b** Enlarged view of the OER curves for S-CoOOH. **c, d** Enlarged view of the OER curves for R-CoOOH.

Moreover, A detailed discussion has been added to the Supplementary Information following Supplementary Fig. S22: Supplementary Fig. S22 presents a magnified view of the OER curves for S-CoOOH and R-CoOOH, derived from Supplementary Fig. S21a. In both samples, two distinct redox peaks are evident. For the redox reactions of CoOOH, we find the peak around 1.0 V should be ascribed to the $\text{Co}^{2+}/2.5^+$ redox process, while the peak around 1.4 V belongs to the $\text{Co}^{2.5+}/3^+$ redox process. These findings were proposed by William C. Chueh et al.²⁰, where they used a suite of correlative operando scanning probe and X-ray microscopy techniques to monitor the transition of local operational chemical, physical and electronic nanoscale structure of single-crystalline $\beta\text{-Co}(\text{OH})_2$ platelet particles under anodic potentials. At pre-catalytic voltages, the pre-catalysts transform into $\alpha\text{-CoO}_2\text{H}_{1.5}\cdot 0.5\text{H}_2\text{O}$, a result of hydroxide intercalation where the oxidation state of cobalt is +2.5. With an increase in voltage to facilitate oxygen evolution, interlayer water and protons are de-intercalated, resulting in the formation of contracted $\beta\text{-CoOOH}$ particles

enriched with Co^{3+} species. Furthermore, in this Nature publication, the researcher corroborated that Co^{3+} (CoOOH) is the active site for the OER, using operando scanning transmission X-ray microscopy (STXM).

12. “ CoLiO_2 ” in Fig. S13b and Table S2 should be denoted as “ LiCoO_2 ”.

Response: We are very grateful for the referee’s comment. We have revised these in the Supplementary Information as below.

Supplementary Fig. S16 (previous Supplementary Fig. S13b) The enlarged results of Co *K*-edge XANES spectra near the adsorption edge.

Supplementary Table S2. FT-EXAFS fitting results of R-CoOOH and S-CoOOH, where CN is the coordination number, σ^2 is the Debye-Waller factor.

Sample	path	CN	σ^2	ΔE_0	R	R-factor
R-CoOOH	Co-O	6	$0.0028 \pm$	1.19 (1.29)	$1.903 \pm$	0.006
			0.0012		0.010	
	Co-Co	6	$0.0044 \pm$	-0.47 (1.27)	$2.848 \pm$	
			0.0009		0.008	

S-CoOOH	Co-O	5.6 ± 0.6	$0.0026 \pm$	1.96 (1.98)	$1.910 \pm$	0.007
			0.0012		0.010	
	Co-Co	4.9 ± 0.6	$0.0049 \pm$	-2.05 (2.1)	$2.839 \pm$	
			0.0010		0.009	
Co(OH) ₂ (II)	Co-O	6.1 ± 0.6	$0.0060 \pm$	1.69 (1.26)	$2.093 \pm$	0.007
			0.0009		0.007	
	Co-Co	6.3 ± 0.7	$0.0080 \pm$	3.52 (1.10)	$3.188 \pm$	
			0.0008		0.007	
LiCoO ₂ (III)	Co-O	4.8 ± 0.3	$0.0024 \pm$	3.01 (0.90)	$1.918 \pm$	0.001
			0.0005		0.004	
	Co-Co	5.7 ± 0.2	$0.0028 \pm$	1.21 (0.46)	$2.817 \pm$	
			0.0003		0.002	

REVIEWER COMMENTS

Reviewer #1 (Remarks to the Author):

The revision is fine for publication.

Reviewer #2 (Remarks to the Author):

The authors have made substantial revisions which address most of the questions. I think this manuscript is almost ready for acceptance. One remaining issue is the magnetic properties or spin state of Co. The authors argue that Co is in high spin state and the materials show ferromagnetic properties. However, according to Fig. R3-6 and Figure 3-7, S-CoOOH shows very low magnetic moment and very low Curie temperature. This should be clarified, because the spin state of S-CoOOH is crucial for the OER activity. Whether the spin at the surface is different from the bulk?

Reviewer #3 (Remarks to the Author):

Since the authors have addressed all the concerns, I would like to recommend publication of this manuscript in Nature Communications.

Response to referees

We thank the three referees for basically agreeing to accept our paper. In response to the comments from the second reviewer, the manuscript has been revised and the changes have been highlighted in the revised version. We feel the quality of the paper has been improved greatly thanks to the input from the referees. Below is a point-by-point response.

Reviewer #1

Remarks to the Author: The revision is fine for publication.

Response: We are sincerely grateful to the reviewer for the time and effort in assessing our manuscript. Your positive remarks and endorsement for publication are greatly appreciated.

Reviewer #2

Remarks to the Author: The authors have made substantial revisions which address most of the questions. I think this manuscript is almost ready for acceptance. One remaining issue is the magnetic properties or spin state of Co. The authors argue that Co is in high spin state and the materials show ferromagnetic properties. However, according to Fig. R3-6 and Figure 3-7, S-CoOOH shows very low magnetic moment and very low Curie temperature. This should be clarified, because the spin state of S-CoOOH is crucial for the OER activity. Whether the spin at the surface is different from the bulk?

Response: We sincerely thank the reviewer for recognizing our work and providing insightful comments.

1) We agree that our S-CoOOH sample has a relatively low magnetic moment.

Here, the effective magnetic moment (μ_{eff}) of cobalt ions was studied by a previously reported method, which provided a systematical investigation of the spin state of cobalt ions in cobalt-based oxides, particularly those exhibiting a mix of low- and high-spin states. Specifically, the Curie-Weiss Law was applied to calculate the effective magnetic moment (μ_{eff}) based on the results of temperature-dependent magnetization (M-T) measurements. (*Nat. Commun.* 2016, 7, 11510; *Angew. Chem. Int. Ed.* 2023, 62, e202216837).

The detailed calculation processes are displayed below. The temperature-dependent magnetizations were measured with a magnetic field of $H = 100$ Oe under field-cooling procedures for S-CoOOH (Fig.R2-1a). The susceptibilities derived from the magnetizations ($\chi = M/H$) obey a Curie-Weiss law: ($\chi = C/(T - T_c)$) where C is Curie constant, and T_c is Curie-Weiss temperature. From the fitting result (Fig.R2-1b), the effective magnetic moment (μ_{eff}) could be calculated through the equation $\mu_{eff} = \sqrt{8C}\mu_B$, where μ_B is Bohr magneton. Based on this method, the calculated effective magnetic moment (μ_{eff}) for S-CoOOH is $0.76 \mu_B$.

Next, the reasons for the low magnetic moment in S-CoOOH are analyzed. Here, we assume that all Co^{3+} ions in CoOOH are in high-spin states. In this case, the number of unpaired electrons n would be 4. The intrinsic magnetic moment could be calculated using the equation: $\mu = \sqrt{n(n+2)}$ where μ represents the

intrinsic magnetic moment, and n denotes the number of unpaired electrons. And the calculated μ is $4.90 \mu_B$. At the same time, when all Co^{3+} in CoOOH are in low-spin states without unpaired electrons, the intrinsic magnetic moment is 0. Following this, the proportion of high-spin state Co^{3+} in our S-CoOOH sample is approximately 15 %. This indicates that the low magnetic moment observed in S-CoOOH is due to the low concentration of high-spin state Co^{3+} .

Fig.R2-1 Magnetic property of S-CoOOH. **a** Temperature-dependent magnetization (M-T) under $H = 100$ Oe. **b** The temperature dependence inverse susceptibility. The dotted line is the fitting result by a Curie-Weiss law.

To identify the low concentration of high-spin state Co^{3+} in our S-CoOOH sample, DFT simulations and Co K -edge FT-EXAFS were employed. When discussing the DFT calculation results, we have shown that the breakage of edge Co-O bonds would lead to the formation of coordinatively unsaturated Co atoms (four-coordinated), exhibiting ferromagnetism properties aligned with high-spin state Co^{3+} . Then, the coordination number (CN) of Co-O bond is fitted based on the Co K -edge FT-EXAFS for both S-CoOOH and R-CoOOH. As shown in Table R2-1, it is revealed that the CN of Co-O for S-CoOOH is ~ 5.6 . Hence, the ratio of 4-coordination Co atoms in S-CoOOH is ~ 20 %, which agrees well with the proportion of high-spin state Co^{3+} (15%) deduced via magnetic analysis. These results clearly indicate that the low magnetic moment observed in S-CoOOH is due to the low concentration of high-spin state Co^{3+} . Yet, the small number of coordinatively unsaturated Co atoms in the high-spin state makes the catalytic property of CoOOH much higher than that of R-CoOOH, demonstrating the superior activity of the high-spin state Co^{3+} atoms. Motivated by this, more efforts could be directed towards increasing the quantity of high-spin state Co^{3+} sites, in order to further enhance the OER activity.

Table R2-1. FT-EXAFS fitting results of R-CoOOH and S-CoOOH, where CN is the coordination number, σ^2 is the Debye-Waller factor.

Sample	path	CN	σ^2	ΔE_0	R	R-factor
R-CoOOH	Co-O	6	0.0028 ±	1.19 (1.29)	1.903 ±	0.006
			0.0012		0.010	
	Co-Co	6	0.0044 ±	-0.47 (1.27)	2.848 ±	
			0.0009		0.008	
S-CoOOH	Co-O	5.6 ± 0.6	0.0026 ±	1.96 (1.98)	1.910 ±	0.007
			0.0012		0.010	
	Co-Co	4.9 ± 0.6	0.0049 ±	-2.05 (2.1)	2.839 ±	
			0.0010		0.009	
Co(OH) ₂ (II)	Co-O	6.1 ± 0.6	0.0060 ±	1.69 (1.26)	2.093 ±	0.007
			0.0009		0.007	
	Co-Co	6.3 ± 0.7	0.0080 ±	3.52 (1.10)	3.188 ±	
			0.0008		0.007	
LiCoO ₂ (III)	Co-O	4.8 ± 0.3	0.0024 ±	3.01 (0.90)	1.918 ±	0.001
			0.0005		0.004	
	Co-Co	5.7 ± 0.2	0.0028 ±	1.21 (0.46)	2.817 ±	
			0.0003		0.002	

2) In our work, the Curie temperature of the S-CoOOH sample is above 300 K. This conclusion is supported by the following evidence:

The Curie point is the critical temperature at which a material undergoes a transition between ferromagnetic and paramagnetic behavior. Below this point, a substance demonstrates ferromagnetic property, while above it, the material becomes paramagnetic characteristic. Fig.R2-2 is the magnetic hysteresis (M-H) loop of S-CoOOH recorded at 300 K. S-CoOOH reflects a hysteresis loop that reaches saturation, leaving a remanent magnetization when the external magnetic field is reduced to zero. These features signify that S-CoOOH

already exhibits ferromagnetic properties at 300 K, thereby confirming that the Curie temperature of S-CoOOH exceeds 300 K.

Fig.R2-2 Magnetic hysteresis (M-H) loop of S-CoOOH recorded at 300 K, where the insets are magnetic ordering patterns of Co³⁺ ions in high-spin states.

3) We agree that the spin states of surface and bulk Co atoms are different. In our work, the term ‘surface’ refers to the ‘edge’ atoms. This point was proposed based on our DFT simulations.

Based on the optimized model of S-CoOOH structure (Fig.R2-3a), the atomic magnetic moments of coordinatively saturated and unsaturated Co atoms are studied (Fig.R2-3b). For R-CoOOH model, non-magnetic properties could be observed for all Co atoms. In contrast, for S-CoOOH model, the coordinatively unsaturated Co atoms at the edge exhibit ferromagnetism property, as evidenced by all calculated atomic magnetic moments aligned in the same direction, characteristic of ferromagnetism. Meanwhile, the fully coordinatively saturated Co atoms in the bulk of S-CoOOH remain non-magnetic.

The emergence of ferromagnetic property in S-CoOOH is due to unpaired electrons in high-spin state Co³⁺ ions. In the low-spin state, Co³⁺ demonstrates a fully paired electron configuration, leading to a complete cancellation of magnetic moments and resulting in a non-magnetic behavior at the ground state. In contrast, the high-spin state Co³⁺ configuration shows unpaired electrons, contributing to a net magnetic moment and exhibiting magnetic properties. Overall, our findings show that in S-CoOOH, the coordinatively unsaturated

Co atoms at the edge are in high-spin states, whereas those coordinatively saturated Co atoms in the bulk are in low-spin states.

Fig.R2-3 **a** Schematic of optimized model of S-CoOOH structure. **b** The distribution of magnetization obtained from DFT, where the inset is *d*-electron configuration of cobalt cations in different spin states at the edge and in the bulk.

To address the comment, we have incorporated the proposed modification into our manuscript (line 136-142, Page 6, highlighted in yellow): Additionally, the effective magnetic moments (μ_{eff}) were calculated using the M-T measurements following a Curie-Weiss Law¹⁶. For R-CoOOH, the calculated μ_{eff} is $0.09 \mu_B$ (Supplementary Fig. S8), which is close to those values reported for low-spin state Co^{3+} ¹⁷. Similarly, the μ_{eff} for S-CoOOH calculated to be $0.76 \mu_B$, suggests the presence of approximately 15 % high-spin state Co^{3+} within the CoOOH structure. This small proportion accounts for the observed low magnetic properties in S-CoOOH, with a detailed discussion provided in Supplementary Fig. S9.

Moreover, we have also added a detailed description of the method used for effective magnetic moment calculation into Methods section of our manuscript (line 387-395, Page 17, highlighted in yellow): **Effective magnetic moment calculation.** The effective magnetic moments (μ_{eff}) of cobalt ions were investigated using temperature-dependent magnetization (M-T) measurements following a Curie-Weiss Law^{16,17}. Here, the M-T measurements were conducted with a magnetic field of $H = 100$ Oe under field-cooling procedures for both R-CoOOH and S-CoOOH. The susceptibilities derived from the magnetizations ($\chi = M/H$) obey a Curie-Weiss law: ($\chi = C/(T - T_c)$ where C is Curie constant, and T_c is Curie-Weiss temperature. The

Curie temperature (C) was extracted from the inverse susceptibility (χ^{-1})-temperature (T) plot, which was fitted based on the M-T measurements data. Then, the effective magnetic moments (μ_{eff}) for both R-CoOOH and S-CoOOH were calculated through the equation $\mu_{eff} = \sqrt{8C}\mu_B$, where μ_B is Bohr magneton.

In addition, a detailed discussion on low magnetic properties in S-CoOOH has been added to the Supplementary Information following Supplementary Fig. S9:

The discussion on low magnetic properties in S-CoOOH

The temperature-dependent magnetizations were measured with a magnetic field of $H = 100$ Oe under field-cooling procedures for S-CoOOH (Supplementary Fig. S9a). From the fitting results (Supplementary Fig. S9b), the calculated μ_{eff} for S-CoOOH is $0.76 \mu_B$.

Next, the reasons for the low magnetic moment in S-CoOOH are analyzed. Here, we assume that all Co^{3+} ions in CoOOH are in high-spin states. In this case, the number of unpaired electrons n would be 4. The intrinsic magnetic moment could be calculated using the equation: $\mu = \sqrt{n(n+2)}$ where μ represents the intrinsic magnetic moment, and n denotes the number of unpaired electrons. And the calculated μ is $4.90 \mu_B$. At the same time, when all Co^{3+} in CoOOH are in low-spin states without unpaired electrons, the intrinsic magnetic moment is 0. Following this, the proportion of high-spin state Co^{3+} in our S-CoOOH sample is approximately 15%. This indicates that the low magnetic moment observed in S-CoOOH is due to the low concentration of high-spin state Co^{3+} .

To identify the low concentration of high-spin state Co^{3+} in our S-CoOOH sample, DFT simulations and Co K -edge FT-EXAFS were employed. When discussing the DFT calculation results, we have shown that the breakage of Co-O bonds would lead to the formation of coordinatively unsaturated Co atoms (four-coordinated), exhibiting ferromagnetism properties aligned with high-spin state Co^{3+} . Then, the coordination number (CN) of Co-O bond is fitted based on the Co K -edge FT-EXAFS for both S-CoOOH and R-CoOOH. As shown in Supplementary Table S2, it is revealed that the CN of Co-O for S-CoOOH is ~ 5.6 . Hence, the ratio of 4-coordination Co atoms in S-CoOOH is $\sim 20\%$, which agrees well with the proportion of high-spin state Co^{3+} (15%) deduced via magnetic analysis. These results clearly indicate that the low magnetic moment in S-CoOOH is due to the low concentration of high-spin state Co^{3+} .

Supplementary Fig. S9 Magnetic property of S-CoOOH. **a** Temperature-dependent magnetization (M-T) under $H = 100$ Oe. **b** The temperature dependence inverse susceptibility. The dotted line is the fitting result by a Curie-Weiss law.

Reviewer #3

Remarks to the Author: Since the authors have addressed all the concerns, I would like to recommend publication of this manuscript in Nature Communications.

Response: We express our profound gratitude to the reviewer for thorough review and valuable feedback on our manuscript. We greatly appreciate your recommendation for publication in Nature Communications. Your support and endorsement of our work are deeply appreciated.